# Uncertainty Estimation for Safety-critical Scene Segmentation via Fine-grained Reward Maximization

**Hongzheng Yang**[1][*]**, Cheng Chen**[2][*]**, Yueyao Chen**[1]**, Markus Scheppach**[3]**, Hon Chi Yip**[1]**, Qi Dou**[1][†]

[1]The Chinese University of Hong Kong
[2]Harvard Medical School & Massachusetts General Hospital
[3] University Hospital of Augsburg

## Abstract

Uncertainty estimation plays an important role for future reliable deployment of deep segmentation models in safety-critical scenarios such as medical applications. However, existing methods for uncertainty estimation have been limited by the lack of explicit guidance for calibrating the prediction risk and model confidence. In this work, we propose a novel fine-grained reward maximization (FGRM) framework, to address uncertainty estimation by directly utilizing an uncertainty metric related reward function with a reinforcement learning based model tuning algorithm. This would benefit the model uncertainty estimation through direct optimization guidance for model calibration. Specifically, our method designs a new uncertainty estimation reward function using the calibration metric, which is maximized to fine-tune an evidential learning pre-trained segmentation model for calibrating prediction risk. Importantly, we innovate an effective fine-grained parameter update scheme, which imposes fine-grained reward-weighting of each network parameter according to the parameter importance quantified by the fisher information matrix. To the best of our knowledge, this is the first work exploring reward optimization for model uncertainty estimation in safety-critical vision tasks. The effectiveness of our method is demonstrated on two large safety-critical surgical scene segmentation datasets under two different uncertainty estimation settings. With real-time one forward pass at inference, our method outperforms state-of-the-art methods by a clear margin on all the calibration metrics of uncertainty estimation, while maintaining a high task accuracy for the segmentation results. Code is available at https://github.com/med-air/FGRM.

## 1  Introduction

Reliable segmentation of safety-critical scene is an important task for a variety of real-world scenarios such as autonomous driving [2, 3, 7, 40] and medical applications [19, 27, 29, 43]. It also plays a fundamental role for higher-level cognitive assistance relying on machine learning, for example, safety-critical soft tissue segmentation is essential for decision-making support towards autonomous robotic surgery [35]. In this regard, tolerance on prediction risk is extremely low, and estimating model uncertainty of the predicted result is equally as important as achieving a high task accuracy itself, especially when the targetted objects exhibit ambiguous boundaries in the scene. Therefore, it is inseparable to provide an estimation of the model uncertainty accompanied with the deterministic network predictions that may introduce risks under safety-critical scenarios.

---

[*]Equal contributions (hzyang22@cse.cuhk.edu.hk, cchen101@mgh.harvard.edu)
[†]Corresponding author (qidou@cuhk.edu.hk)

37th Conference on Neural Information Processing Systems (NeurIPS 2023).

Although uncertainty estimation of deep learning models for image segmentation has been intensively studied [1, 14], almost all previous methods solely resort to task objectives (e.g., cross-entropy, IoU or Dice) to train the network, and then investigate different inference strategies on the network, such as sampling multiple predictions to compute variance [24, 25, 26, 32], implicitly estimating uncertainty through feature density [12, 38], and designing evidence-based or deterministic layers to predict uncertainty [5, 6, 36]. Despite effectiveness to a certain extent, these practices commonly suffer from the issue of the absence of the uncertainty estimation metric to be considered, neither in the model training stage nor during the estimation process. This leads to indirect calibration of model confidence and unclear prior, which would affect the quality of estimated uncertainty, especially in the presence of ambiguous regions or out-of-distribution cases. Recently, reinforcement learning (RL) for model tuning has emerged with new achievements, such as improving language models from human feedback [33] and enhancing vision models with task rewards [28, 34]. We are inspired that the uncertainty estimation task can also be tackled with a RL reward maximization paradigm by innovating an uncertainty metric related reward function. This would have advantages over previous methods, because the model can be effectively fine-tuned with explicit guidance from the model confidence for calibrating the prediction risks.

The key challenges for achieving reward optimization-based uncertainty estimation lie in how to design the reward, and more importantly, how to update the network parameters during reward maximization. According to the latest works on vision tasks with RL [34, 42], the policy gradient methods estimate reward-weighted gradients for parameter updates, and the reward function is not necessarily differentiable. However, in these studies, the reward is uniformly weighted for all the network parameters, which may not be optimal because the influence of each model parameter on the reward function could differ from one to another. In addition, it can be expected that optimizing a network for dense segmentation prediction with feedback from a single reward could be challenging. In these regards, this process demands a meticulously crafted mechanism for parameter updates to identify the optimal solution within a constrained exploration space. Therefore, we consider that a designed fine-grained parameter update is required for model tuning process.

In this paper, we propose a novel **f**ine-**g**rained **r**eward **m**aximization (FGRM) framework for uncertainty estimation in safety-critical scene segmentation. Specifically, we design a reward function that is directly related to uncertainty estimation, such as the calibration metrics, and demonstrate that by maximizing the reward function with reinforcement learning algorithms, the segmentation model can be explicitly tuned and well-calibrated for both segmentation prediction and uncertainty estimation. Importantly, to deal with the crucial yet challenging parameter updates with reward function, we leverage fisher information matrix, which carries the importance information of each parameter to network outputs, and yields a new fine-grained reward-weighting scheme for gradients of each network parameter. Moreover, we leverage the evidential deep learning framework to pre-train segmentation backbone, which makes the two types of uncertainties, i.e., aleatoric and epistemic uncertainty, to be separately estimated for reward maximization. Our main contributions are summarized as follows:

- We propose a new reward optimization paradigm for uncertainty estimation, which can explicitly provide guidance to calibrate prediction risk and model confidence by designing a reward function that is closely tied to the uncertainty estimation.

- We design an effective fine-grained parameter update mechanism for reward maximization, which imposes fine-grained reward-weighting for each network parameter and enables more effective model tuning based on the reward function.

- We conduct extensive experiments on two medical datasets of safety-critical applications, including laparoscopic cholecystectomy scene segmentation and endoscopic submucosal dissection scene segmentation. Our new method consistently outperforms different types of state-of-the-art uncertainty estimation methods across all evaluation metrics.

## 2 Related Work

**Uncertainty estimation for image segmentation.** For the image segmentation task, related works on uncertainty estimation can be categorized into probabilistic methods [4, 11, 13, 39], model ensemble [24, 25, 26, 32], evidence-based methods [5, 6, 36], deep deterministic methods [12, 38] and auxiliary network-based methods [9, 20]. In specific, probabilistic methods estimate model uncertainty via dropout [13] and conditional variational autoencoders [4, 11, 39]. The ensemble-

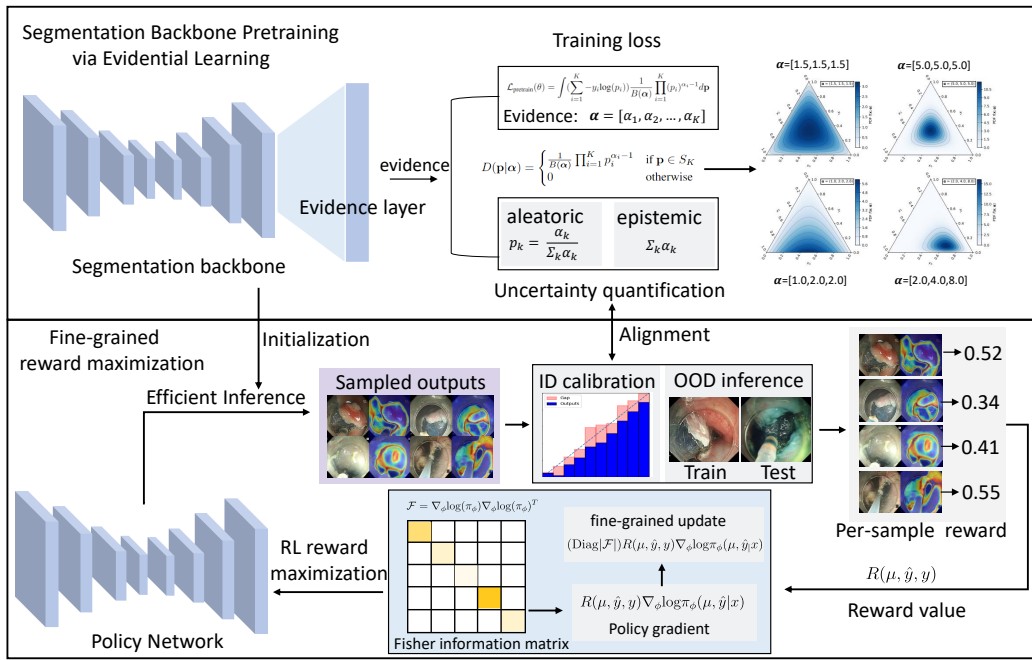

**Figure 1:** The overview of our proposed fine-grained reward maximization (FGRM) framework for uncertainty estimation. The policy network is initialized with the parameters of a segmentation backbone, which is pre-trained with evidential learning to generate estimation for aleatoric and epistemic uncertainty separately. The policy network is tuned with a designed uncertainty estimation reward function by using a fine-grained parameter update scheme.

based methods hold the insight to train a number of models for uncertainty estimation, and use stochastic segmentation networks [24, 32] to model a joint distribution over the entire map to reduce computation burdens. The evidence-based methods incorporate evidential layers into the network to model the conjugate prior of softmax prediction [31], which aims to directly predict the uncertainty of segmentation results. The deep deterministic methods can have low computational cost by exploiting the implicit feature density when the model architectures satisfy the Bi-lipschitz regularization [12]. Auxiliary network-based method designs an auxiliary branch to learn the confidence criterion, such as the True Class Probability (TCP). It requires an additional network branch for confidence estimation, and is often limited by the class imbalance problem, with a huge number of high-confidence samples during training. Despite much success previously achieved, these methods all resort to the model trained on task objectives, without taking into account the uncertainty estimation metric during learning. Our proposed method differs from them significantly, by exploring reinforcement learning algorithm to update the base segmentation model w.r.t. to a designed uncertainty reward.

**Reinforcement learning for vision tasks.** Using reinforcement learning for computer vision tasks is not completely new [28]. To improve the accuracy of scene segmentation, early methods use RL algorithms to iteratively refine the segmentation output [30], sequentially attend to various regions of the image [8], and realize multi-step interactive segmentation [37]. Very recently, there are some works exploring how to tune a base vision model to optimize the task-specific reward on different tasks, where the reward was designed as the task-specific evaluation metrics [34]. For panoptic segmentation, they adopt the per-sample panoptic quality [22] as the reward. For object detection task, they tune the model to optimize the recall and mean average precision metric. For tasks where the reward/metric are difficult to express by hand-crafted formulations such as the visual editing task, people propose to harness human feedback as instructions for reinforcement learning from human feedback (RLHF) [42]. The method needs human annotators to rank different outputs of the base model and then train a reward model that reflects human preferences. All these works demonstrated impressive performance on their respective tasks, and imply a promising trend to tackle vision tasks with reinforcement learning based model tuning. To the best of our knowledge, there is still no related work on addressing model uncertainty estimation with such a new RL-based paradigm.

# 3 Proposed Method

An overview of our proposed method is illustrated in Figure.1. Our method aims to learn a scene segmentation model capable of delivering not only highly accurate predictions but also high-quality uncertainty estimation for safety-critical scenes. We consider two types of uncertainty, i.e., aleatoric uncertainty and epistemic uncertainty, with aleatoric uncertainty accounting for irreducible data uncertainty, such as ambiguous tissue boundaries in surgical images, while epistemic uncertainty resulting from the lack of sufficient knowledge about unseen data. Let $X$ denote the input image space and $Y$ denote the label mask space, without loss of generality, we construct a differentiable segmentation model $\pi_{\hat{\theta}}$ with model parameters $\hat{\theta}$, which maps the input image $x$ to the segmentation prediction and the prediction uncertainty. The learned segmentation model $\pi_{\hat{\theta}}$ is required to yield the estimations of aleatoric uncertainty for in-distribution (ID) data and epistemic uncertainty for out-of-distribution (OOD) data during real-time model inference.

## 3.1 Maximizing Uncertainty Estimation Reward with Reinforcement Learning

For existing uncertainty estimation approaches, a segmentation network is typically optimized by maximum likelihood estimation (MLE) based on the task objective function such as cross-entropy or Dice loss. However, in such a model training process, there is no explicit guidance to calibrate the prediction risk and model confidence. The challenge roots in the fact that there is usually no ground truth annotation for learning prediction uncertainty, thus it is hard to formulate a suitable optimization objective. To tackle this problem, we propose to make use of the reinforcement learning algorithm which explicitly maximizes a reward function that reflects uncertainty estimation metric, to tune the pre-trained segmentation network to calibrate the model parameters for both segmentation prediction and uncertainty estimation.

Specifically, we propose a policy network $\pi_{\phi}$ with the same architecture of a pre-trained segmentation model $\pi_{\hat{\theta}}$. The details of the segmentation model pre-training will be described in Sec. 3.3. The policy network's parameter $\phi$ is initialized with $\hat{\theta}$, then tuned to maximize a reward function $R(\mu, \hat{y}, y)$, where $\mu, \hat{y}, y$ denote the uncertainty estimation, segmentation prediction and ground truth, respectively. We propose to design the reward function as calibration metrics that can quantify how closely the uncertainty is aligned with the model prediction risk, including both the in-distribution prediction confidence miscalibration and over-confidence in the out-of-distribution samples at inference. For the confidence miscalibration risk in in-distribution samples, the reward can be designed to minimize the difference in expectation between prediction accuracy and estimated aleatoric uncertainty. For the overconfidence risk in out-of-distribution samples, the reward can be designed to quantify whether the model is able to estimate higher epistemic uncertainty to OOD samples compared to their ID counterparts. The detailed formulation of the reward function is provided in the supplementary material. With the designed reward function, the RL objective can be expressed as:

$$\mathcal{J}(\phi) = \mathbb{E}_{(x,y)}[R(\mu, \hat{y}, y) - \beta \log(\frac{\pi_{\phi}(x)}{\pi_{\hat{\theta}}(x)})], \tag{1}$$

where the first term is the reward maximization and the second term is a pixel-wise Kullback–Leibler (KL) penalty. The goal of the KL term is to mitigate over-optimization towards the reward and prevent policy network parameters $\phi$ deviating too much from the pre-trained segmentation model $\hat{\theta}$. The hyper-parameter $\beta$ controls the strength of the KL penalty.

In this regard, the policy network $\pi_{\phi}$ can be updated to maximize the objective $\mathcal{J}(\phi)$. In particular, we desire the parameter $\phi$ update can be fine-grained, and start with an easy-to-optimize initialization, i.e., the update should be specific for different parameters and the base model pretraining could provide a reasonable initialization for RL training. Moreover, the base model should have the ability to distinguish between the different types of uncertainty and ensure efficient sampling for reward computation. In the following, we will describe the design of RL-based parameter tuning and the base model pre-training in detail.

## 3.2 Fine-grained Parameter Update

To maximize the objective $\mathcal{J}(\phi)$, the main challenge is how to estimate the gradient of the reward term in Eq. 1. According to the policy gradient theorem [18], the gradient can be estimated as

$\nabla_\phi \mathbb{E}_{(x,y)}[R(\mu, \hat{y}, y)] = \mathbb{E}_{(x,y)}[R(\mu, \hat{y}, y)\nabla_\phi \log \pi_\phi(\mu, \hat{y}|x)]$, in which the gradient of each parameter is weighted by the reward $R(\mu, \hat{y}, y)$ uniformly. We propose that the update for model $\pi_\phi$ should not be identical everywhere in the parameter space. This is because that the parameters of $\pi_\phi$ may contribute differently to the uncertainty estimation as well as segmentation prediction, therefore they can influence the reward function differently. With this insight, the uniform update scheme may not enable effective exploration for maximizing the uncertainty estimation reward. Furthermore, the dealing task here is quite challenging since we are optimizing a network to conduct dense prediction with feedback from the function of a single reward. Given the constrained exploration, together with guidance from the sparse reward value, finding a solution that achieves uncertainty estimation aligned with model risk is difficult.

To address this challenge, we propose to tune model parameters with uncertainty reward in a fine-grained manner. In other words, we aim to weigh the reward for each network parameter individually. Our hypothesis is that since the parameters of policy network $\pi_\phi$ need to keep relatively close to those of the pre-trained segmentation model $\pi_{\hat{\theta}}$, this constraint limits the exploration space during RL training. If we can assign larger update weights to those parameters that significantly influence the uncertainty reward within this constrained exploration space, it can more efficiently reach the optimal solution for maximizing rewards. Inspired by elastic weight consolidation scheme in continual learning [23], we resort to the fisher information matrix, which carries information about the importance of each parameter to the model outputs. We compute the fisher information matrix as the covariance of the gradient of log likelihood as follows:

$$\text{Diag}|\mathcal{F}| = \text{Diag}(\mathbf{1} \oslash \nabla_\phi \log(\pi_\phi)\nabla_\phi \log(\pi_\phi)^T). \tag{2}$$

The calculation for fisher information matrix requires only the first-order derivative which can be computation efficient for large models. Next, we use the diagonal components in fisher information matrix to identify which parameters are more important and re-weight the parameter update during the RL-based model tuning. The updating scheme becomes as:

$$\phi = \phi + \eta(\sum_s ((\text{Diag}|\mathcal{F}|)R(\mu, \hat{y}, y)\nabla_\phi \log \pi_\phi(\mu_s, \hat{y}_s|x)) - \beta \nabla_\phi \log(\frac{\pi_\phi(x)}{\pi_{\hat{\theta}}(x)})), \tag{3}$$

where $s$ denotes each pixel. In this equation, the gradient of log likelihood for each parameter is weighted by both the reward value and diagonal element of the fisher information matrix. Larger value in the elements indicates a greater update step for network parameters, and such a gradient is computed for each individual pixel. The sum of pixel-wise gradients will be used to update the parameters. In this way, the parameter update is weighted by the information it carries about the final prediction, enabling more effective model tuning in the constrained parameter space which cannot deviate too much from the pre-trained parameters.

### 3.3 Segmentation Backbone Pre-training via Evidential Learning

To achieve effective reward maximization of uncertainty estimation, pre-training of the segmentation backbone also plays an important role, because it provides the capability to distinguish different uncertainty types that need to be well aligned with different sources of prediction risks. However, in a standard segmentation model, the final layer is often a softmax layer, which can only provide a point estimate of the predicted class probabilities. Their probability values produced by the softmax layer mix the aleatoric and epistemic uncertainty together, making it unclear to differentiate between the two types of uncertainty.

To address this issue, we borrow the idea of evidential deep learning [5] to train the model for explicitly parameterizing the conjugate prior of the categorical distribution. The segmentation of input $x$ is modeled as a Dirichlet distribution $D(\mathbf{p}|\boldsymbol{\alpha})$, where $\mathbf{p}$ is a simplex $S_K = \{\mathbf{p}| \sum_{k=1}^K p_k = 1, p_k \geq 0\}$ representing the categorical probabilities with $K$ segmentation classes. In this regard, the probability density function of Dirichlet distribution is given by:

$$D(\mathbf{p}|\boldsymbol{\alpha}) = \begin{cases} \frac{1}{B(\boldsymbol{\alpha})} \prod_{i=1}^K p_i^{\alpha_i - 1} & \text{if } \mathbf{p} \in S_K, \\ 0 & \text{otherwise}, \end{cases} \tag{4}$$

where $B(\boldsymbol{\alpha})$ is the multinomial beta function as a normalizing factor. During the training process, the model iteratively measure the amount of support from data, i.e. evidence, in favor of the sample to be

---

**Algorithm 1** FGRM algorithm

---

**Input**: segmentation backbone pre-training epoches $E_s$, RL training epoches $E_r$, reward maximization learning rate $\eta$, reward function $R(\mu, \hat{y}, y)$.
**Output**: model with reliable uncertainty estimation results
**Segmentation Backbone Pre-training:**
1:  Initialize segmentation model $\pi_\theta$
2:  **for** $e = 0, 1, \dots, E_s$ **do**
3:      sample batch of training data (**x, y**) from the training set
4:      calculate $L_{\text{pretrain}}(\theta)$                                            ▷ Eq.(5)
5:      update parameters: $\theta \leftarrow \theta - \eta\nabla_\theta L_{\text{pretrain}}(\theta)$
6:  **end for**
7:  **return** pretrained parameters $\hat{\theta}$
**RL Reward Maximization:**
1:  initialize policy network $\pi_\phi$ with pre-trained parameters $\hat{\theta}$
2:  **for** $e = 0, 1, \dots, E_r$ **do**
3:      sample batch of training data (**x, y**) from the validation set
4:      $\text{Diag}|\mathcal{F}| = \text{Diag}(\mathbf{1} \oslash \nabla_\phi\log(\pi_\phi)\nabla_\phi\log(\pi_\phi)^T)$          ▷ Eq.(2)
5:      $\phi \leftarrow \phi + \eta(\sum_s((\text{Diag}|\mathcal{F}|)R(\mu, \hat{y}, y)\nabla_\phi\log\pi_\phi(\mu_s, \hat{y}_s|x)) - \beta\nabla_\phi\log(\frac{\pi_\phi(x)}{\pi_{\hat{\theta}}(x)}))$   ▷ Eq.(3)
6:  **end for**
7:  **return** tuned policy network $\pi_\phi$ with reliable uncertainty estimation

---

classified into a certain class. By replacing the last softmax layer with a non-negative evidence layer, we can represent the Dirichlet parameters $\boldsymbol{\alpha}$ with the model output $\pi_{\hat{\theta}}(x) + 1$. In this way, the model is parameterized a distribution over the output categorical distribution. The aleatoric uncertainty can be quantified as expected categorical probability value $\alpha_k/\sum_k(\alpha_k)$ and the epistemic uncertainty can be quantified as the summation of Dirichlet distribution parameter $\sum_k(\alpha_k)$. For maximum likelihood training, we treat the $D(\mathbf{p}|\boldsymbol{\alpha})$ as the prior over the log likelihood $\log(\text{Categorical}(y|p))$ and calculate the Bayes maximum likelihood loss by integrating out $\mathbf{p}$ as:

$$L_{\text{pretrain}}(\theta) = \int (\sum_{i=1}^{K} -y_i\log(p_i))\frac{1}{B(\boldsymbol{\alpha})}\prod_{i=1}^{K}(p_i)^{\alpha_i - 1}d\mathbf{p}. \tag{5}$$

The model optimized by $L_{\text{pretrain}}$ could provide a proper initialization $\hat{\theta} \in \text{argmin}L_{\text{pretrain}}(\theta)$ for RL training. At inference time, we can get the Dirichlet parameters with one forward pass of $\pi_{\hat{\theta}}$, thus quantify the reward and uncertainty efficiently.

### 3.4   Implementation Details

Overall, the algorithm of our entire framework is presented in Algorithm 1. In our implementation, we employ an adapted TransUNet as segmentation backbone. We replace the last softmax layer with a non-negative evidence layer. The evidence layer is implemented by the softplus function. For the base model pre-training, we use the Adam optimizer, with learning rate initialized to 1e-4. We totally trained 10 epoches on the training set, with batch size 4. For the maximization of uncertainty estimation reward, we tune the base model to maximize the reward on a held-out validation set. For the ID miscalibration risk, we design the reward as the negative logarithm ECE function [15] $-ln(\text{ECE})$. For the OOD over-confidence risk, we generate a corrupted version of the original validation and test set following [12, 16]. The reward was calculated as the uncertainty values ratio of the corrupted OOD samples compared to their ID counterparts. The model was updated by the policy gradient with our proposed fine-grained parameter update scheme. The learning rate and batch size was initialized as 1e-4 and 4, respectively.

## 4   Experiments

### 4.1   Experimental Datasets and Evaluation Metrics

We validate our method on two datasets of safety-critical surgical scene segmentations, i.e., laparoscopic cholecystectomy (LC) scene segmentation and endoscopic submucosal dissection (ESD)

**Table 1:** Comparison with state-of-the-art methods on laparoscopic cholecystectomy dataset for safety-critical tissue segmentation. Results are reported with mean±std of three independent runs.

| Method | In-distribution Calibration | | | OOD Inference | | Runtime |
| --- | --- | --- | --- | --- | --- | --- |
| | ECE ↓ | MI ↑ | DICE ↑ | PR ↑ | BR ↑ | (ms) ↓ |
| Segmentation Backbone | 19.46±1.12 | 2.88±0.76 | 71.22±1.22 | 0.88±0.33 | 0.08±0.04 | **0.052** |
| Deep Ensemble [26] | 16.20±2.01 | 3.82±0.86 | 72.59±1.44 | 0.92±0.23 | 0.12±0.04 | 0.201 |
| Layer Ensemble [25] | 17.04±0.97 | 3.94±0.59 | 72.23±1.04 | 1.00±0.25 | 0.16±0.07 | 0.072 |
| DUM [38] | 17.95±1.31 | 4.31±0.69 | 70.68±0.64 | 1.20±0.47 | 0.27±0.03 | 0.055 |
| LDU [12] | 14.42±0.72 | 4.93±0.45 | 73.13±1.57 | 1.44±0.44 | 0.35±0.09 | 0.054 |
| MC-dropout [13] | 17.27±0.60 | 3.31±0.49 | 72.42±1.57 | 1.03±0.22 | 0.15±0.06 | 0.510 |
| LHU [21] | 17.14±1.95 | 3.41±0.81 | 72.57±1.94 | 1.16±0.37 | 0.12±0.04 | 0.054 |
| NatPN [6] | 11.74±0.66 | 4.79±0.41 | 74.16±0.97 | 1.56±0.33 | 0.31±0.03 | **0.052** |
| ConfidNet [9] | 16.76±1.19 | 4.58±0.70 | 71.32±1.24 | 1.41±0.66 | 0.32 ±0.13 | 0.057 |
| Auxiliary feat. [20] | 16.87±1.92 | 4.35±0.75 | 71.32±1.24 | 1.33±0.55 | 0.28±0.11 | 0.055 |
| FGRM (Ours) | **9.63±0.74** | **5.87±0.47** | **74.88±0.91** | **1.85±0.25** | **0.47±0.02** | **0.052** |

**Table 2:** Comparison with state-of-the-art methods on endoscopic submucosal dissection dataset for safety-critical tissue segmentation. Results are reported with mean±std of three independent runs.

| Method | In-distribution Calibration | | | OOD Inference | | Runtime |
| --- | --- | --- | --- | --- | --- | --- |
| | ECE ↓ | MI ↑ | DICE ↑ | PR ↑ | BR ↑ | (ms) ↓ |
| Segmentation Backbone | 17.76±0.31 | 2.08±0.19 | 84.12±0.43 | 0.82±0.22 | 0.10±0.03 | **0.046** |
| Deep Ensemble [26] | 15.22±0.44 | 3.21±0.22 | 85.62±0.62 | 0.96±0.17 | 0.15±0.04 | 0.192 |
| Layer Ensemble [25] | 15.78±0.37 | 3.29±0.12 | 85.23±0.23 | 1.09±0.11 | 0.13±0.06 | 0.060 |
| DUM [38] | 16.07±0.54 | 3.51±0.27 | 83.11±0.85 | 1.14±0.10 | 0.15±0.07 | 0.047 |
| LDU [12] | 14.12±0.58 | 3.66±0.53 | 84.85±0.50 | 1.33±0.12 | 0.24±0.05 | **0.046** |
| MC-dropout [13] | 16.65±0.42 | 3.11±0.40 | 83.24±0.77 | 1.05±0.04 | 0.18±0.03 | 0.417 |
| LHU [21] | 16.62±0.62 | 3.36±0.34 | 83.31±1.22 | 1.12±0.31 | 0.14±0.04 | 0.047 |
| NatPN [6] | 13.67±0.91 | 4.16±0.53 | 86.77±0.93 | 1.45±0.22 | 0.32±0.07 | **0.046** |
| ConfidNet [9] | 15.52±1.12 | 3.53±0.65 | 84.41±0.61 | 1.19±0.25 | 0.26 ±0.10 | 0.048 |
| Auxiliary feat. [20] | 15.76±1.15 | 3.41±0.61 | 84.41±0.88 | 1.14±0.40 | 0.18 ±0.08 | 0.047 |
| FGRM (Ours) | **10.42±0.88** | **4.72±0.25** | **87.23±0.54** | **1.78±0.19** | **0.54±0.03** | **0.046** |

scene segmentation. **Dataset-1:** For LC segmentation dataset, we adopt the public dataset Cholec-Seg8K [17], which contains 8,080 laparoscopic cholecystectomy image frames extracted from 17 video clips. Annotations are provided for segmentation of four different soft tissues, including abdominal wall, liver, gastrointestinal tract, and fat. **Dataset-2:** For ESD segmentation dataset, we collected a dataset with 1,203 image frames from 30 endoscopic surgical videos. The dataset was annotated by expert surgeons for the submucosal tissue, mucosa tissue, muscle tissue, and blood vessel. For data pre-processing, all the images were resized to $256 \times 256$ and normalized with zero mean and unit variance as the network inputs. For each dataset, we first randomly split 20% data for a held-out testing, and further split 80% of remaining data for training and 20% for validation. To demonstrate the broad applicability of our method, we also provide experimental evaluations on Cityscape [10] dataset for urban scene segmentation in Appendix A.5.

We evaluate our method under two uncertainty estimation scenarios, including in-distribution calibration and out-of-distribution inference. The inclusion of out-of-distribution inference aims to comprehensively assess the uncertainty estimation of our model when encountering data that deviates from their training distribution. The uncertainty estimation of in-distribution samples is conducted directly on the held-out test data, since there is no significant distribution shift between the training data and test data. The uncertainty estimation of out-of-distribution samples is conducted by generating a corrupted version of test data following [12, 16] and involving surgical video data of different organs. We adopt three commonly-used metrics for in-distribution calibration, including Expected Calibration Error (ECE) [15], Uncertainty-error mutual information (MI) [19] and Dice score, and two metrics for out-of-distribution inference, including the Pixel Ratio (PR) and Box Ratio (BR) [41].

## 4.2 Comparison with State-of-the-art Methods

**Comparison state-of-the-art methods.** Our method is compared with current state-of-the-art uncertainty estimation approaches, including ensemble-based methods of deep ensemble (**DE**) [26] and layer ensemble (**LE**) [25], deterministic uncertainty estimation methods of **DUM** [38] and **LDU** [12], probabilistic method of **MC-dropout** [13] and **LHU** [21], evidence-based method of **NatPN** [6] and auxiliary network based method of **ConfidNet** [9] and **Auxiliary feat.** [20]. In particular, **DE** simply uses an ensemble of neural networks trained independently to quantify the predictive uncertainty. **LE** lowers the computation cost of DE by ensembling from a single network's different depths. **DUM** trains a model with Bi-Lipschitz constraints and exploits the feature space densities to quantify the uncertainty. **LDU** advances scalable DUM by relaxing the Bi-Lipschitz constraints with a distinction maximization layer. **MC-dropout** approximates the Bayesian predictive uncertainty with a number of dropout runs. **LHU** designed the network consisted of two branches, which predicted the mean and variance separately. **NatPN** uses a normalizing flow to account for the epistemic uncertainty and proposes an input-dependent evidence update rule for the posterior distribution parameters. **ConfidNet** uses an additional network branch to learn the prediction confidence (True Class Probability) on a held-out validation set. **Auxiliary feat.** cascaded several consecutive convolution layers after the feature extractor network to learn to identify the segmentation errors. We also compare with a baseline model, which denotes training our **Segmentation Backbone** without using the evidential learning or any uncertainty estimation method.

**Experimental results.** The Table 1 shows the quantitative results on LC segmentation. We can see that different uncertainty estimation methods can generally improve the uncertainty estimation over baseline for both settings of the ID calibration and OOD inference. Our method achieves superior performance than the comparison methods on all the evaluation metrics and scenarios. Compared with the method NatPN, our approach obtains improvement of 2.1% in ECE and 0.29% in PR. Compared with MC-dropout that requires multiple forward runs to obtain uncertainty estimation, our method not only achieves higher uncertainty estimation quality, but also requires less inference time, which is important for real-time intra-operative healthcare applications. Table 2 shows the results on ESD segmentation. Similarly, compared to other methods, our approach achieves efficient uncertainty estimation and obtains higher performance on all evaluation metrics, with improvement of 3.25% on ECE and 0.83% on PR. The superiority of our method is attributed to the explicit model tuning with uncertainty estimation reward and the fine-grained parameter updates scheme that enables efficient parameter exploration of policy network. The visual results in Fig. 2(a) show that our method is able to produce varying degrees of uncertainty estimation according to the potential prediction risk. Besides the in-distribution result visualization, we also conducted additional tests on ESD surgical data involving out-of-distribution organs distinct from those covered by training data. Specifically, we applied the model trained on ESD surgical data from the stomach, to assess on ESD surgical data from the esophagus and rectum. Figure 2(b) showcases the results of our uncertainty estimation for these OOD cases, demonstrating the effectiveness of our method in providing meaningful uncertainty estimates when confronted with different organ types.

## 4.3 Ablation Analysis of the Proposed Method

We conduct experiments to investigate several key properties in our method: **i)** contribution of each key component; **ii)** effect of fine-grained parameter update; **iii)** relation between estimated uncertainty and prediction correctness; and **iv)** progression of uncertainty estimation and segmentation prediction.

**Contribution of each component.** We first validate the three key components in our method, i.e., uncertainty estimation reward maximization, fine-grained parameter update, and evidential learning, by continuously adding each component one by one to the plain MLE model. When evidential learning is not employed, the maximum class probability generated by the softmax function is used as both the aleatoric and epistemic uncertainty. As shown in Fig. 3 (a), each component plays an important role in improving the quality of uncertainty estimation on the two datasets. This demonstrates the benefits of using reinforcement learning with fine-grained parameter update and the evidential learning-based pre-training on improving uncertainty estimation.

**Effect of fine-grained parameter update.** We analyze the effect of fine-grained parameter update by varying the strengths of KL penalty, that is the hyperparameter $\beta$ in Eq. 3. As observed in Fig. 3 (b), with the proposed fine-grained parameter update scheme, our method consistently obtains better uncertainty estimation performance under different strengths of KL penalty. Also we can see that

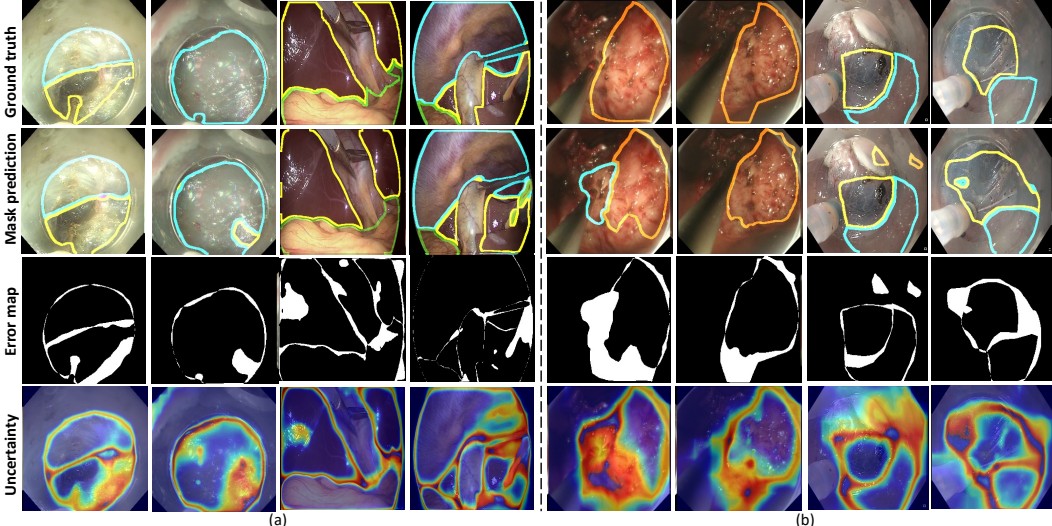

**Figure 2: a)** Estimated uncertainty map using our method on endoscopic submucosal dissection (ESD) of stomach dataset (first two columns) and laparoscopic cholecystectomy dataset (last two columns) for in-distribution calibration; **b)** Estimated uncertainty map using our method on ESD of esophagus (first two columns) and ESD of rectum (last two columns) for different organs calibration.

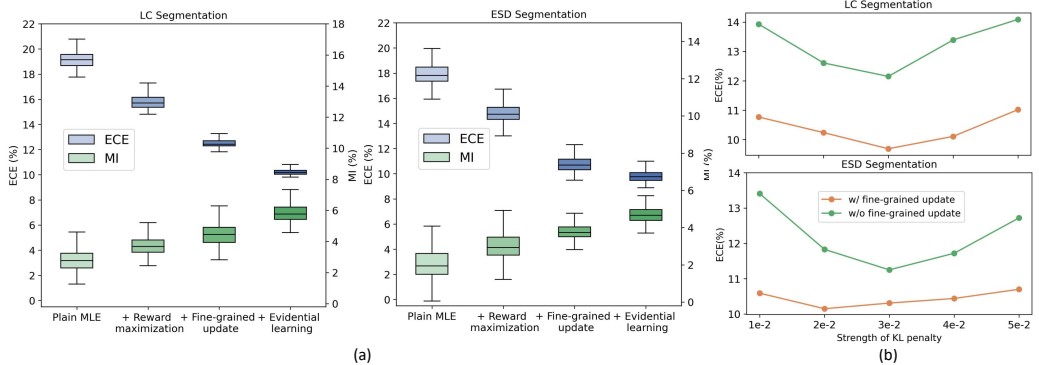

**Figure 3: a)** Effect of the three key components in our method, i.e., reward maximization, fine-grained parameter update, and evidential learning; **b)** Change of uncertainty estimation performance with the strength of KL penalty.

with the fine-grained parameter update, the uncertainty estimation is more stable with respect to different strengths of KL penalty. Since the hyperparameter $\beta$ controls the constrains on parameter optimization space during RL tuning, these results demonstrate that the fine-grained parameter update scheme contributes to effectively finding a better solution in the constrained parameter space.

**Relation between estimated uncertainty and prediction correctness.** We show the estimated uncertainty with the prediction correctness in Fig. 4 (a). We can see that the "Plain MLE" model produces low uncertainty values for both incorrect and correct predictions, showing that the model is miscalibrated. With our FGRM method, the estimated uncertainty values are clearly related to the prediction correctness, that is the model is able to give high uncertainty values for incorrect predictions and low uncertainty values for correct predictions.

**Progression of uncertainty estimation and segmentation prediction.** We plot the ECE and dice curve during the reward maximization process to see their progression. As shown in Fig. 4 (b), we observe the model uncertainty calibration and Dice performance quickly increase during the reward tuning on the validation set and stay stable around 2k steps. This shows that our reward maximization improves the uncertainty estimation quality, as well as the accuracy of segmentation predictions.

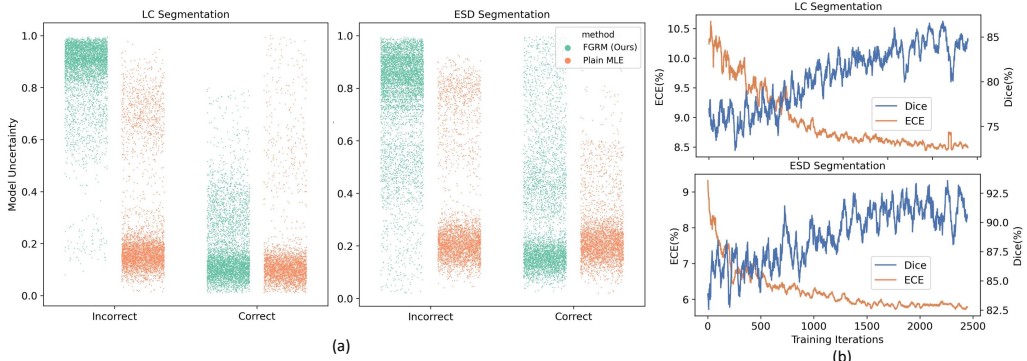

**Figure 4: a)** Scatter plot of pixel-wise uncertainty with prediction correctness. **b)** Progression of ECE and Dice curves during reward maximization process.

## 5   Conclusion

We have proposed a novel reward optimization-based uncertainty estimation framework for safety-critical scene segmentation. By maximizing a uncertainty estimation reward with a meticulously crafted fine-grained parameter update scheme, our framework provides explicit guidance to calibrate the prediction risk and model confidence with efficient network tuning. The superior efficacy of our method is demonstrated on two safety-critical surgical scene segmentation tasks, with both improved uncertainty estimation quality and segmentation predictions.

**Limitations and future work.** Since the calibration metrics for the uncertainty estimation in in-distribution samples and out-of-distribution samples are different, our framework currently designed different reward functions to deal with the two scenarios. In the future, we would like to explore how to design a common reward function for model tuning such that a unified model can deliver high-quality uncertainty estimation under different situations. Additionally, we are planning to delve into more advanced RL techniques in order to reduce the frequency of reward calls and model updates. This will further enhance the training efficiency of our method.

**Social Impact.** Our FGRM framework provides a promising solution for efficiently estimating the uncertainty of model predictions, which is crucial for various real-world applications such as medical diagnosis or interventions. By producing high-quality uncertainty estimation, we can ensure the safer deployment of predictive models in practical applications.

## Acknowledgements

This work was supported in part by Hong Kong Innovation and Technology Commission Project No. ITS/237/21FP, in part by Hong Kong Research Grants Council Project No. T45-401/22-N, and in part by Science, Technology and Innovation Commission of Shenzhen Municipality Project No. SGDX20220530111201008.

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
