# A Appendix

In this appendix, we begin by providing detailed information on the reward design for our FGRM method. Subsequently, we discuss the reasons why maximizing rewards via policy gradients can be sub-optimal. In addition, we present additional qualitative and quantitative ablation studies showing results on OOD data and uncertainty map visualization. Finally, we present potential of our proposed framework on urban scene segmentation for autonomous driving applications.

## A.1 Reward Design

For the in-distribution (ID) calibration, we adopt the Expected Calibration Error (ECE) metric as the reward to maximize, which assesses whether more confident (i.e. less uncertain) predictions are more likely to be correct. It quantifies the difference between confidence and accuracy on average. For the calculation, we divide the prediction confidences into $M$ bins and calculate the average confidence and accuracy for each bin. Then, we compute the weighted average of the difference between accuracy and confidence across all the bins using the following formula:

$$\text{ECE} = \sum_{m=1}^{M} \frac{|N_m|}{N} |\text{acc}(N_m) - \text{conf}(N_m)| \tag{1}$$

where $N$ is the number of predictions, $N_m$ is the predictions that fall into the m-th bin. To increase the range of ECE values, we employ the negative logarithm of ECE as the final reward. By maximizing the reward of ECE metric, the confidence miscalibration can be reduced. For out of distribution (OOD) inference, it is desired that the model can assign high epistemic uncertainty to the OOD regions compared to their ID counterparts. In this regards, we design the FGRM to maximize the ratio between uncertainty of OOD regions and ID regions. It can be calculated as,

$$\frac{\sum_{s \in \text{OOD}} \sigma_s}{\sum_{s \in \text{ID}} \sigma_s} \tag{2}$$

where $s$ represents the pixel index and $\sigma_s$ represents the estimated epistemic uncertainty value. By maximizing this designed reward, the model learns to mitigate the OOD over confidence.

## A.2 Policy Gradient based Reward Maximization for Segmentation Backbone

Directly adopting the policy gradient method to optimize the segmentation backbone might be sub-optimal. Considering the optimization objective,

$$\mathcal{J}(\phi) = \mathbb{E}_{(x,y)}[R(\mu, \hat{y}, y) - \beta \log(\frac{\pi_\phi(x)}{\pi_{\hat{\theta}}(x)})], \tag{3}$$

where $\pi_\phi$ denotes the policy network we want to optimize and $R(\mu, \hat{y}, y)$ is the reward function. The first term is the reward maximization and the second term is a KL penalty to mitigate over-optimization towards the reward. The gradient with resepect to the first term is $R(\mu, \hat{y}, y)\nabla_\phi \log \pi_\phi(\mu, \hat{y}|x)$. The log-likelihood gradient is uniformly weighted by the reward value. Also note that updating parameters $\phi$ based on policy gradient method would converge to the solution $\pi_\phi^* \in \{\pi_\phi | \nabla_{\pi_\phi} \mathcal{J}(\phi) = 0\}$. Differentiating objective $\mathcal{J}(\phi)$ with respect to $\pi_\phi$, we can express the closed-form solution as

$$\pi_\phi^* = \frac{\pi_\theta \exp(R(\boldsymbol{\mu}^i, \hat{\boldsymbol{y}}^i, \boldsymbol{y}^i)/\beta)}{\int \pi_\theta \exp(R(\boldsymbol{\mu}^i, \hat{\boldsymbol{y}}^i, \boldsymbol{y}^i)/\beta)d(\boldsymbol{x}^i, \boldsymbol{y}^i)} \tag{4}$$

The solution $\pi_\phi^*$ can be viewed as weighting the MLE model $\pi_\theta$ by the normalized exponential reward isotropically. The exploration space for optimizing the segmentation backbone is constrained, leading to sub-optimal solutions. Our fine-grained parameter update mechanism addresses this issue by distributing rewards individually to each network parameter based on its impact on the uncertainty reward. This approach enables us to efficiently achieve the optimal solution for reward maximization.

## A.3 Uncertainty Map Visualization

In Fig. 1, we present visualization of uncertainty maps and corresponding predictions on the LC and ESD datasets. Regarding LC, all methods struggle to accurately segment the tract region

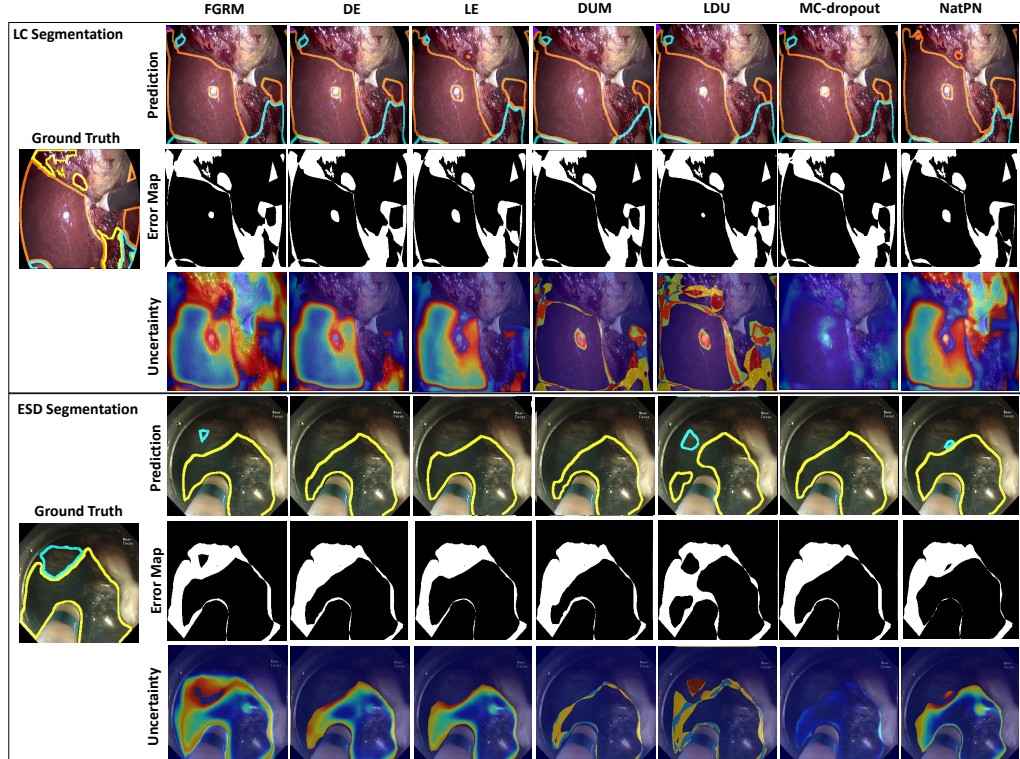

**Figure 1:** Visualization of estimated uncertainty on LC and ESD datasets. For LC (first three rows), we present the liver, tract, and cystic duct boundaries using orange, yellow, and cyan-blue color respectively. For ESD (last three rows), we represent the submucoal and muscle boundaries by yellow and cyan-blue color respectively. Uncertainty scores are denoted by colors, with blue indicating low uncertainty and red indicating high uncertainty.

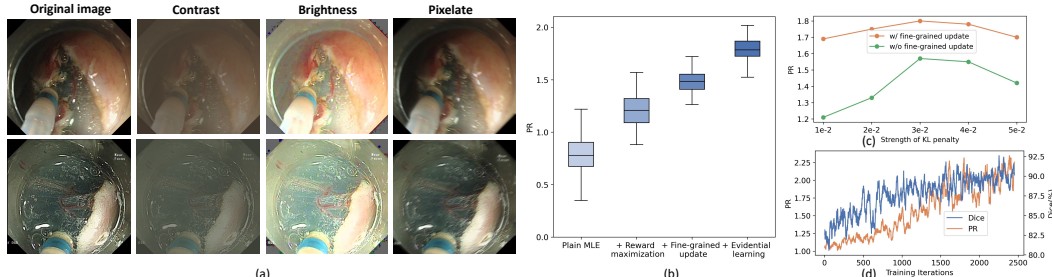

**Figure 2:** a) Examples of generated out-of-distribution (OOD) images with perturbations; b-d) ablation studies for OOD inference with endoscopic submucosal dissection (ESD) dataset on the three key components, the change of uncertainty estimation with the strength of KL penalty, and the progression of uncertainty metric and Dice curves during reward maximization process respectively.

due to reflection and bleeding. Our method successfully generates calibrated uncertainty for the incorrectly segmented regions, providing valuable insights into the areas where the segmentation may be unreliable. For ESD, it is a challenging in distinguishing between the ambiguous submucosa and muscle tissues. Our method is able to generate uncertainties that specifically highlight potential errors in the segmentation prediction for those ambiguous regions.

## A.4 Additional Results on OOD Data

We present some examples of generated OOD examples in Fig. 2(a). We randomly applied four types of perturbations, including contrast, brightness, pixelate, and noise, to the original images. We can

see that those perturbations mainly adjust the contrast, brightness, and clarity of surgical images, which might be encountered in practical scenarios. We conducted additional ablation studies on the OOD scenario using the endoscopic submucosal dissection dataset, including contribution of each key component, effect of the fine-grained parameter update, and the progression of uncertainty estimation and segmentation prediction. The results are presented in Fig. 2(b)-(d). These results on the OOD scenario further confirm the findings observed in the ID scenario.

### A.5 Proposed Framework for Autonomous Driving Applications

To demonstrate the versatility of our approach across various applications, we conducted additional evaluation on the Cityscapes [2] dataset, which presents a diverse and realistic collection of urban scene images from 50 different cities. This dataset is created for semantic segmentation with 5,000 frames annotated with high-precision pixel-level segmentation masks involving 30 fine-grained classes.

In Table 1, we present the results of our uncertainty estimation framework when applied to the Cityscapes dataset. Our method consistently outperforms all the comparison approaches. This consistent superiority confirms the broad potential and applicability of our approach, particularly in scenarios where safety and precision are important, such as in the field of autonomous driving.

**Table 1:** Comparison with different methods on Cityscapes dataset for urban scene segmentation.

| Method | In-distribution Calibration | | | OOD Inference | | Runtime |
|---|---|---|---|---|---|---|
| | ECE ↓ | MI ↑ | DICE ↑ | PR ↑ | BR ↑ | (ms) ↓ |
| Segmentation Backbone | 11.94 | 3.51 | 78.15 | 0.70 | 0.06 | 0.063 |
| Deep Ensemble [6] | 10.66 | 4.91 | 79.10 | 1.37 | 0.26 | 0.334 |
| Layer Ensemble [5] | 10.40 | 4.73 | 79.04 | 1.15 | 0.11 | 0.089 |
| LDU [3] | 9.14 | 5.75 | 78.69 | 1.55 | 0.35 | 0.064 |
| MC-dropout [4] | 11.15 | 4.09 | 78.46 | 1.23 | 0.19 | 0.751 |
| NatPN [1] | 9.32 | 5.30 | 78.45 | 1.62 | 0.34 | 0.071 |
| FGRM (Ours) | 8.52 | 6.97 | 79.23 | 1.77 | 0.48 | 0.064 |