# OpenReview forum: "Uncertainty Estimation for Safety-critical Scene Segmentation via Fine-grained Reward Maximization"
_NeurIPS.cc/2023/Conference — NeurIPS 2023 poster_

### Official Review · Reviewer_iwgk · 2023-06-29

**Soundness:** 3 good
**Presentation:** 3 good
**Contribution:** 3 good
**Rating:** 6
**Confidence:** 3

**Summary:**

As existing approaches for uncertainty estimation have been limited by the guidance for calibrating the prediction risk and model confidence, the paper proposes a novel fine-grained reward maximization (FGRM) framework, which addresses uncertainty estimation by reinforcement learning based model tuning with an uncertainty metric related reward function. It adopts the fisher information matrix for capturing parameter importance, acting as weights for fine-grained updates. Besides, evidential pre-training is incorporated to distinguish between aleatoric and epistemic uncertainty. The experimental results on two surgical datasets show FGRM improves uncertainty estimation for both ID and OOD data while not harming original segmentation performance.

**Strengths:**

1. The paper is well motivated for the uncertainty estimation problem, with reasonable application of fisher information matrix and evidential learning. Though built upon previous works that uses RL for guided training, I think the fine-grained update mechanism is novel and the contributions are clear.
2. The presentation is clear and easy to follow.
3. The empirical performance is great comparing to baselines.

**Weaknesses:**

1. While enjoying superior empirical performance, it would be better to provide some theoretical insights for FGRM like the uncertainty bounds, which is significant in the uncertainty estimation context.
2. The experiment section has some ambiguous points (see limitations below).

**Questions:**

1. In line 306, why you say `without harming the accuracy` instead of `while promoting the accuracy` with Dice performance increases during reward tuning?
2. As you use evidential learning to distinguish between aleatoric and epistemic uncertainty, what is the principle behind how it benefits your uncertainty estimation? You haven't explained this clearly in the ablation study.
3. What does `efficient sampling` mean in the lower part of Figure 1? How does your sample algorithm differs from simple mini-batch sampling?
4. Typo in line 324: safter => safer.

**Limitations:**

1. The paper lacks theoretical provements for FGRM like the uncertainty bounds, which is significant in the uncertainty estimation context.
2. In Figure 3 and Figure 4(b), the paper doesn't record FGRM's performance on the OOD counterpart, which is an important contribution claimed in the paper.

---

> ### Author Rebuttal · Authors · 2023-08-10
>
> Thank you for your positive feedback regarding the novelty of our proposed fine-grained parameter update scheme and the great empirical performance of our method. Our responses to your comments are as follows.
>
> > * While enjoying superior empirical performance, it would be better to provide some theoretical insights for FGRM like the uncertainty bounds, which is significant in the uncertainty estimation context.
>
> Reply: Thank you for the comment. We would like to provide theoretical insights regarding our proposed fine-grained parameter update scheme. For uncertainty metric reward maximization, directly adopting the policy gradient method can be sub-optimal, which can be proofed with the closed-form solution of the policy network. In RL tuning process, the overall optimization objective can be expressed as, $  J(\phi) =  E_{({x}, {y})}[R({\mu}, {\hat{y}}, {y}) - \beta \text{log}(\frac{\pi_{\phi}({x})}{\pi_{\hat{\theta}}({x})})] $. According to the policy gradient theorem, the gradient with respect to the first term can be calculated as $R(\mu, \hat{y}, y) \nabla_{\phi} \text{log} \pi_{\phi}(\mu, \hat{y}|x)$. Updating parameters  $\phi$ based on the policy gradient is towards to $\pi_{\phi}^* \in \{ \pi_{\phi} | \nabla_{\pi_{\phi}}\mathcal{J(\phi)} = 0 \}$, which can be expressed as, $\pi_{\phi}^* = \frac{\pi_{\theta}\text{exp}(R(\boldsymbol{\mu}^i, \boldsymbol{\hat{y}}^i, \boldsymbol{y}^i)/\beta)}{\int \pi_{\theta}\text{exp}(R(\boldsymbol{\mu}^i, \boldsymbol{\hat{y}}^i, \boldsymbol{y}^i)/\beta) d(\boldsymbol{x}^i, \boldsymbol{y}^i)}$. The solution $\pi_{\phi}^*$ can be viewed as weighting the MLE model by the normalized exponential reward. The limited space for exploration constrains the optimization, resulting in sub-optimal solutions. Our proposed fine grained update mechanism tackles this problem by assigning rewards separately to each network parameter according to its influence on the uncertainty reward.
>
> > * In line 306, why you say without harming the accuracy instead of while promoting the accuracy with Dice performance increases during reward tuning?
>
> Reply: Thank you for the comment. The choice of phrasing is intended to highlight that our method aims to improve uncertainty estimation without negatively impacting the accuracy of the segmentation predictions. This emphasizes that our primary focus is on enhancing uncertainty estimation. However, it is also valid to use “while promoting accuracy” as evidenced by the increasing Dice performance during reward tuning.
>
> > * As you use evidential learning to distinguish between aleatoric and epistemic uncertainty, what is the principle behind how it benefits your uncertainty estimation? You haven't explained this clearly in the ablation study.
>
> Reply: The principle behind how evidential learning benefits our uncertainty estimation lies in its capability to provide a better understanding of uncertainty types. Specifically, aleatoric uncertainty emerges from the inherent, irreducible variability presented in the data itself, such as when dealing with unclear tissue boundaries in surgical images. differently, epistemic uncertainty arises due to limited knowledge about unseen data. By being able to differentiate these two types of uncertainty, our model gains the capacity to provide more informative uncertainty estimates. In the ablation study, we further added a model named “Vanilla EDL” that exclusively uses evidential learning without the incorporation of the fine-grained RL algorithm. This ablation experiment aims to investigate the contribution of evidential learning by directly comparing it with the “Vanilla MLE” model. The results, presented in the table below, clearly indicate that evidential learning enhances uncertainty estimation by enabling the model to capture different types of uncertainty more accurately. This helps the model to better understand the limits of its predictions and produce more meaningful uncertainty estimates.
>
> **LC segmentation**
>
> |           |       | ID calibration |        | OOD inference |      |
> |-----------|-------|----------------|--------|---------------|------|
> |   Method | ECE ↓ |      MI ↑      | Dice ↑ |      PR ↑     | BR ↑|
> | Vanilla MLE | 19.46 |     2.88      |  71.22 |      0.88     | 0.08 |
> |    Vanilla EDL   |  12.98 |      4.20      |  73.66 |      1.45     | 0.21 |
> |    FGRM   |  9.63 |      5.87      |  74.88 |      1.85     | 0.47 |
>
> **ESD segmentation**
>
> |           |       | ID calibration |        | OOD inference |      |
> |-----------|-------|----------------|--------|---------------|------|
> |   Method  | ECE ↓ |      MI ↑      | Dice ↑ |      PR ↑     | BR ↑ |
> | Vanilla MLE | 17.76 |     2.08      |  84.12 |      0.82     | 0.10 |
> |    Vanilla EDL   |  12.15 |      3.68      |  86.20 |      1.40     | 0.18 |
> |    FGRM   |  10.42 |      4.72      |  87.23 |      1.78     | 0.54 |
>
> > * What does efficient sampling mean in the lower part of Figure 1? How does your sample algorithm differs from simple mini-batch sampling?
>
> Reply: The term “efficient sampling” in Fig. 1 refers to the inference process of uncertainty estimation. Our method only requires a single forward pass to obtain the uncertainty prediction, which is highly efficient compared to other ensemble or probabilistic-based methods that require multiple forward passes for the uncertainty estimation.
>
> > * In Figure 3 and Figure 4(b), the paper doesn't record FGRM's performance on the OOD counterpart, which is an important contribution claimed in the paper.
>
> Reply:  For Fig. 3 and Fig. 4(b), we further added ablation studies on the OOD scenario using the endoscopic submucosal dissection dataset, including contribution of each key component, effect of the fine-grained parameter update, and the progression of uncertainty estimation and segmentation prediction. The results are presented in Fig. 3(b)-(d) of the uploaded PDF file. These results on the OOD scenario reaffirm the findings observed in the ID scenario.

---

> > ### Comment · Reviewer_iwgk · 2023-08-19
> > **Thank you**
> >
> > Thank you for the effort during the rebuttal, I will keep the positive assessment of the paper.

---

### Official Review · Reviewer_jxwz · 2023-07-06

**Soundness:** 2 fair
**Presentation:** 3 good
**Contribution:** 2 fair
**Rating:** 4
**Confidence:** 4

**Summary:**

This paper introduces a novel Fine-Grained Reward Maximization (FGRM) framework to improve uncertainty estimation in deep segmentation models for safety-critical applications. The approach uses a reinforcement learning-based model tuning paradigm to optimize and calibrate the model. The FGRM framework is the first to leverage reinforcement learning for uncertainty estimation in safety-critical vision tasks, demonstrating improved performance on two surgical scene segmentation datasets.

**Strengths:**

1. The paper is well-written, logically organized, and effectively explains the novel aspects of the proposed framework.
2. The method has been rigorously tested on two large safety-critical surgical scene segmentation datasets, demonstrating superior performance.

**Weaknesses:**

1. The evaluation metrics Uncertainty error mutual information (MI),  Pixel Ratio (PR), and Box Ratio (BR) can not be found in  [18].
2. In the related works and experiments, lack of discussion and comparison with the auxiliary network-based method, e.g. [18] and Corbiere, Charles, et al. "Confidence estimation via auxiliary models."

**Questions:**

See the weaknesses.

**Limitations:**

While the paper is largely comprehensive, there are some areas where it falls short:

1. Comparison with Other Methods: The paper claims superiority over state-of-the-art methods, however, it lacks a direct comparison or in-depth discussion with recent popular general uncertainty estimation methods, such as "Confidence estimation via auxiliary models" or methods specifically tailored for medical image segmentation like [18]. Including these comparisons would validate their claims more convincingly.

2. Metric Citation Issue: There seems to be a mistake in the metric citation, which raises concerns about the overall quality of the experimental setup.

---

> ### Author Rebuttal · Authors · 2023-08-10
>
> We appreciate the reviewer's valuable comments, providing us with opportunities for improvement and clarification. We would like to address each of your comments in detail as follows.
>
> > * Comparison with other methods: In the related works and experiments, lack of discussion and comparison with the auxiliary network-based method, e.g. [18] and Corbiere, Charles, et al. "Confidence estimation via auxiliary models."
>
> Reply: Following your suggestion, we have added the raised auxiliary network-based method ConfidNet (i.e., Corbiere Charles et al [R1]) into our experiments for comparison. The tables below show the performance of our FGRM method and ConfidNet on two surgical video datasets as well as a urban scene dataset of Cityscape (an additional general dataset). The results show that our method achieves better uncertainty estimation performance compared to Corbiere Charles et al. We will include these additioanl experiments in final version.
>
> **LC segmentation dataset**
>
> |           |       | ID calibration |        | OOD inference |      |
> |-----------|-------|----------------|--------|---------------|------|
> |   Method&nbsp;&nbsp;&nbsp;&nbsp;&nbsp;&nbsp;  | ECE ↓ |      &nbsp;&nbsp;&nbsp;&nbsp;&nbsp;&nbsp;&nbsp;&nbsp;&nbsp;&nbsp;MI ↑      | Dice ↑&nbsp;&nbsp;&nbsp;&nbsp;&nbsp;&nbsp; |      PR ↑     | BR ↑ &nbsp;&nbsp;&nbsp;&nbsp;&nbsp;&nbsp;|
> | ConfidNet | 16.76 |      &nbsp;&nbsp;&nbsp;&nbsp;&nbsp;&nbsp;&nbsp;&nbsp;&nbsp;&nbsp;4.58      |  71.32 |      1.41     | 0.32 |
> |    FGRM (ours)   |  9.63 |      &nbsp;&nbsp;&nbsp;&nbsp;&nbsp;&nbsp;&nbsp;&nbsp;&nbsp;&nbsp;5.87      |  74.88 |      1.85     | 0.47 |
>
> **ESD segmentation dataset**
>
> |           |       | ID calibration |        | OOD inference |      |
> |-----------|-------|----------------|--------|---------------|------|
> |   Method&nbsp;&nbsp;&nbsp;&nbsp;&nbsp;&nbsp;  | ECE ↓ |      &nbsp;&nbsp;&nbsp;&nbsp;&nbsp;&nbsp;&nbsp;&nbsp;&nbsp;&nbsp;MI ↑      | Dice ↑&nbsp;&nbsp;&nbsp;&nbsp;&nbsp;&nbsp; |      PR ↑     | BR ↑ &nbsp;&nbsp;&nbsp;&nbsp;&nbsp;&nbsp;|
> | ConfidNet | 15.52 |      &nbsp;&nbsp;&nbsp;&nbsp;&nbsp;&nbsp;&nbsp;&nbsp;&nbsp;&nbsp;3.53      |  84.41 |      1.19     | 0.26 |
> |    FGRM (ours)   |  10.42 |      &nbsp;&nbsp;&nbsp;&nbsp;&nbsp;&nbsp;&nbsp;&nbsp;&nbsp;&nbsp;4.72      |  87.23 |      1.78     | 0.54 |
>
> **Cityscapes dataset**
>
> |           |       | ID calibration |        | OOD inference |      |
> |-----------|-------|----------------|--------|---------------|------|
> |   Method&nbsp;&nbsp;&nbsp;&nbsp;&nbsp;&nbsp;  | ECE ↓ |      &nbsp;&nbsp;&nbsp;&nbsp;&nbsp;&nbsp;&nbsp;&nbsp;&nbsp;&nbsp;MI ↑      | Dice ↑&nbsp;&nbsp;&nbsp;&nbsp;&nbsp;&nbsp; |      PR ↑     | BR ↑ &nbsp;&nbsp;&nbsp;&nbsp;&nbsp;&nbsp;|
> | ConfidNet | 10.14 |      &nbsp;&nbsp;&nbsp;&nbsp;&nbsp;&nbsp;&nbsp;&nbsp;&nbsp;&nbsp;4.69      |  78.15 |      1.18     | 0.19 |
> |    FGRM (ours)   |  8.52 |      &nbsp;&nbsp;&nbsp;&nbsp;&nbsp;&nbsp;&nbsp;&nbsp;&nbsp;&nbsp;6.97      |  79.23 |      1.77     | 0.48 |
>
> > * The evaluation metrics Uncertainty error mutual information (MI), Pixel Ratio (PR), and Box Ratio (BR) can not be found in [18].
>
> Reply: Thank you for the comment. In paper [18], a comprehensive benchmark of various uncertainty estimation methods are conducted. The metric of uncertainty-error overlap in [18] is a discrete version of the uncertainty error mutual information used in our experiment. We agree the importance of adding explicit references to the papers that thoroughly describe these metrics. Hence, we would like to add reference [R2] for the metric of uncertainty error mutual information and reference [R3] for the metrics of pixel ratio and box ratio.
>
> Thank you very much for the careful review, it definitely makes our paper more rigorous. But we also would like to assure the reviewer regarding the quality and soundness of our experimental setup. There is no need to worry, and we will release all our code, data and models upon publication of the paper.
>
> References:
>
> [R1] Corbiere, C., Thome, N., Saporta, A., Vu, T.H., Cord, M. and Perez, P. Confidence estimation via auxiliary models. IEEE Transactions on Pattern Analysis and Machine Intelligence, 44(10), 2021, pp.6043-6055.
>
> [R2] Judge, T., Bernard, O., Porumb, M., Chartsias, A., Beqiri, A. and Jodoin, P.M. Crisp-reliable uncertainty estimation for medical image segmentation. In International Conference on Medical Image Computing and Computer-Assisted Intervention. 2022, pp. 492-502.
>
> [R3] Zepf, K., Wanna, S., Miani, M., Moore, J., Frellsen, J., Hauberg, S., Feragen, A. and Warburg, F. Laplacian Segmentation Networks: Improved Epistemic Uncertainty from Spatial Aleatoric Uncertainty. arXiv preprint arXiv:2303.13123. 2023.

---

> > ### Comment · Reviewer_jxwz · 2023-08-19
> >
> > Thank you for your response. After considering your reply, I still have reservations regarding the comprehensiveness of your experimental evaluations:
> >
> > Detailed Discussion and Comparisons: I noticed a lack of a comprehensive and detailed discussion when comparing your method to other recent state-of-the-art (SOTA) methods for general uncertainty estimation, such as the auxiliary network-based method. A deeper dive into the pros, cons, and unique features of each method would enhance the quality of your comparison.
> >
> > Medical Imaging Comparisons: I'd like to emphasize the importance of comparing your approach with methods tailored specifically for medical image segmentation, such as [17] and [18]. These works are benchmarks in the field, and their omission from the comparison is noticeable.
> >
> > Evaluation Metrics References: While you've referenced the evaluation metrics from [17], [18], and [R3], I observed that there's no direct comparison made with these methods. Given that [17] and [18] are recognized as the latest SOTA for medical image segmentation, such comparisons would be highly relevant and beneficial.
> >
> > Due to these concerns, I'm inclined to think that the experimental design might have certain biases and lacks comprehensiveness. Consequently, I feel it's appropriate to adjust my rating for this submission.

---

> > ### Author Response · Authors · 2023-08-19
> > **Thank You and Our Further Response**
> >
> > Thank you very much for taking time to read our response. We aim to address your remaining concerns as follows.
> >
> > *Detailed Discussion and Comparisons*: Our paper has compared with a range of SOTA approaches, including probabilistic method [11], model ensemble-based methods [22][23], deep deterministic methods [10][35], and evidence-based method [6]. We have also included a comparison with an auxiliary network-based method ConfidNet in our response. We acknowledge the importance of a detailed discussion for in-depth comparison, but space limits prevented us from including such analysis in our original submission. Following your suggestion, we hereby provide a detailed analysis of pros and cons for each type of method. We will include them into the final version to enhance the writing quality of our comparisons.
> >
> > The probabilistic method MC Dropout [11] is simple in implementation using dropout layers, but it depends on multiple forward runs for uncertainty estimation. The model ensemble-based method Deep Ensembles [23] combines the outputs of multiple models for reliable uncertainty estimation, but the training of multiple models has computation burdens. Layer Ensemble [22] attaches multiple heads to intermediate layers of a network, achieving efficient uncertainty estimation with a single forward pass. Deep deterministic methods [35][10] quantify uncertainty through geometrical or statistical properties of hidden features, providing accurate out-of-distribution uncertainty estimation, but their Bi-lipschitz regularization can be unstable in deeper models. The evidence-based method [6] distinguishes aleatoric and epistemic uncertainty, but due to the lack of evidence ground truth, the model can only be trained with observed one-hot labels, which may have the tendency to peak the second-order distribution. ConfidNet designs an auxiliary network to learn a novel confidence criterion, making it applicable to any pre-trained segmentation model, albeit requiring an additional network for confidence estimation.
> >
> > Despite the notable achievements in these methods, they share one common limitation, i.e., relying on models trained on task objectives without considering the uncertainty estimation metric during the learning process. The superior performance of our method can be attributed to the following key factors. Firstly, our method explicitly optimizes uncertainty estimation metrics via a reward function, thereby directly calibrating prediction risk and model confidence. Furthermore, our fine-grained parameter updates scheme enables the effective parameter exploration of the policy network. In addition, our incorporated evidential learning layer allows us to provide more informative estimates of different uncertainty types. As shown in Table 1&2 in our paper, our method also has the advantage of less inference time, a critical factor for real-time intra-operative healthcare applications. The limitations of our method include the need for an additional RL reward maximization process and the current use of two reward functions for different types of uncertainty.
> >
> > *Medical Imaging Comparison*: For our comparison with auxiliary network-based methods in our rebuttal, we focus on ConfidNet (TPAMI’21) because it is a more recent work than [18] (MICCAI’19). Given that ConfidNet and [18] share similar insights by leveraging an auxiliary network, we only added ConfidNet in our comparison due to limited rebuttal time. However, we understand your point that [18] is a more relevant work as it is tailored for medical image segmentation. We are implementing [18] now and will post the results asap.
> >
> > Meanwhile, we would like to bring your attention to Table 1&2 in our paper, where we have included a MICCAI 2022 paper with the method of Layer Ensemble [22] for comparison. It is exactly a recent SOTA uncertainty estimation method designed for medical image segmentation. Our proposed method outperforms this MICCAI'22 paper by a large margin. Hope this can help relieve your concern.
> >
> > *Evaluation Metrics References*: Thank you for your further comments on direct comparison on the references. We agree that it is helpful to improve the paper, and actively try to do it. For the comparison with [17], since their code is not released, we tried our best to re-implement their method based on the information provided in their paper. Unfortunately, it is unstable in learning the joint latent space of input images and corresponding segmentation maps. As a result, the obtained uncertainty estimation was unsatisfactory, even presenting lower performance than baseline. We acknowledge the importance of including more SOTA medical imaging methods into comparison, so we are currently conducting experiments on [18] and [R3] to make our experimental evaluation more comprehensive. We believe our comparison with various types of general methods as well as dedicated medical imaging methods will effectively demonstrate the effectiveness of our method.

---

> > > ### Author Response · Authors · 2023-08-20
> > > **Comparison with More Medical Imaging Methods**
> > >
> > > Thank you for your patience. We have finished the implementation of the auxiliary feat. and auxiliary segm. networks from [18] based on their publicly released code. The tables below present the quantitative comparison of our FGRM method with the auxiliary feat. [18] on LC and ESD segmentation datasets. We compare with the results of the auxiliary feat. in the tables since auxiliary feat. obtains slightly better performance than auxiliary segm. on the two datasets, even though their performances are largely comparable. We can see that our method consistently outperforms [18] across all evaluation metrics. The superior performance of our method demonstrates the benefits of explicit model tuning with uncertainty estimation metrics-based RL algorithm. We will include the comparison with [18] into the final version to provide a more comprehensive analysis against uncertainty estimation methods tailored for medical image segmentation.
> > >
> > > **LC segmentation dataset**
> > > |            |        | ID calibration |         | &nbsp;&nbsp;&nbsp;&nbsp;&nbsp;&nbsp;&nbsp;&nbsp;&nbsp;&nbsp;&nbsp;&nbsp;&nbsp;&nbsp;&nbsp;&nbsp;&nbsp;&nbsp;OOD inference |       |
> > > |:---------:|:-----:|:--------------:|:------:|:-------------:|:----:|
> > > | Method | ECE ↓ | &nbsp;&nbsp;MI ↑ | Dice ↑ | PR ↑ | BR ↑ |
> > > | Auxiliary feat.| 16.87 | &nbsp;&nbsp;4.35 | 71.32 | 1.33 | 0.28 |
> > > | FGRM | 9.63 | &nbsp;&nbsp;5.87 | 74.88 | 1.85 | 0.47 |
> > >
> > > **ESD segmentation dataset**
> > > |            |        | ID calibration |         | &nbsp;&nbsp;&nbsp;&nbsp;&nbsp;&nbsp;&nbsp;&nbsp;&nbsp;&nbsp;&nbsp;&nbsp;&nbsp;&nbsp;&nbsp;&nbsp;&nbsp;&nbsp;OOD inference |       |
> > > |:---------:|:-----:|:--------------:|:------:|:-------------:|:----:|
> > > | Method | ECE ↓ | &nbsp;&nbsp;MI ↑ | Dice ↑ | PR ↑ | BR ↑ |
> > > | Auxiliary feat.| 15.76 | &nbsp;&nbsp;3.41 | 84.41 | 1.14 | 0.18 |
> > > | FGRM | 10.42 | &nbsp;&nbsp;4.72 | 87.23 | 1.78 | 0.54 |

---

### Official Review · Reviewer_WcWT · 2023-07-09

**Soundness:** 4 excellent
**Presentation:** 3 good
**Contribution:** 4 excellent
**Rating:** 8
**Confidence:** 3

**Summary:**

This paper proposes a novel method for uncertainty estimation. A segmentation network is first pre-trained by considering a generative model where the segmentation of an input $x$ is drawn from a Dirichlet distribution, which enables MLE.

The main contribution of the paper is then the reinforcement learning (RL) algorithm proposed whereby a novel reward function is used to maximise uncertainty estimation. Lastly, the authors posit that not all parameters in the network should be updated accordingly and use ideas from EWC in continual learning to learn parameter-specific update rules.

To learn aleatoric and epistemic uncertainty quickly, the authors assume a generative model where the segmentation is drawn from a Dirichlet distribution, thus optimising the MLE through integration of the conjugate prior and the likelihood function. The aleatoric and epistemic (calibrated) uncertainties can consequently be obtained through the learned $\alpha$ parameter.

**Strengths:**

* This is an excellent paper. It is well written, clear in its intentions. The methodology is clearly presented (Figure 1 is excellent), it is clear how the algorithm works and why each section of the method was developed. The results compared to various baselines help consolidate the strength of the paper.

* There are many novelties in this paper, which in isolation might already be interesting contributions. Taken as a whole, the authors have presented a compelling piece of work. The idea to combine i) evidential learning (parameterising the segmentation as $s ~ D(p|\alpha)$ for uncertainty ii) using EWC to learn parameter updates necessary in the RL algorithm for calibrating uncertainty and iii) the RL reward function  are interesting and novel. The results are subsequently impressive.

**Weaknesses:**

* I would have liked to have seen evidence of the calibration on a more diverse set of datasets to better show the applicability and performance of the algorithm. A diverse set such as Cityscapes (real scenes), a dataset such as BraTS (tumour segmentation from MRI scans) in addition to the surgical videos used in this paper would have strengthened the work.

**Questions:**

* Did the authors consider also papers such as those which propose learning heteroscedastic uncertainty with MC dropout as a baseline such as in Kendal et al.(https://arxiv.org/abs/1703.04977). This method is used quite significantly and it would have been nice to see how it performs in comparison.

* It would be nice to see uncertainty maps of other models to compare. Is this possible?

* It is not entirely clear how the RL algorithm mitigates confidence miscalibration and OOD over confidence. This is only really mentioned in passing in Section 3.4 in the implementation details for Equation 1

* In the experiments, did the authors consider a model where only evidential learning was used as another baseline to compare against vanilla MLE? It would be nice to understand how strong of a baseline that is and whether this could be used in isolation.

---

> ### Author Rebuttal · Authors · 2023-08-10
>
> We sincerely appreciate your insightful comments and positive feedback regarding the recognition of “many novelties” in our paper, clear presentation of our methodology and strength of experimental evaluation. We would like to provide the necessary clarifications and improvements in response to your comments as follows.
>
>  > * Evidence of the calibration on a more diverse set of datasets to better show the applicability and performance of the algorithm.
>
> Reply: Thank you for your comment. Following your suggestion, we have conducted additional experiments on the Cityscapes dataset to evaluate the uncertainty estimation of segmentation on real urban scenes. The results are presented below, along with comparisons to various baselines. It is evident from the results that our method outperforms all the comparison methods for uncertainty estimation on the Cityscapes dataset. These results demonstrate the effectiveness of our method in model uncertainty calibration on diverse datasets, including both surgical videos in medical applications and urban scenes in natural images.
>
> **Cityscapes dataset**
>
> |           |       | ID calibration |        | OOD inference |      |
> |-----------|-------|----------------|--------|---------------|------|
> |   Method  | ECE ↓ |      &nbsp;&nbsp;MI ↑      | Dice ↑&nbsp;&nbsp;&nbsp;&nbsp;&nbsp;&nbsp; |      PR ↑     | BR ↑ |
> | Backbone | 11.94 |      &nbsp;&nbsp;3.51      |  78.15 |      0.70     | 0.06 |
> |    LayerEnsemble   |  10.40 |      &nbsp;&nbsp;4.73      |  79.04 |      1.15     | 0.11 |
> |    DeepEnsemble   |  10.66 |      &nbsp;&nbsp;4.91      |  79.10 |      1.37     | 0.26 |
> |    LDU   |  9.14 |      &nbsp;&nbsp;5.75      |  78.69 |      1.55     | 0.35 |
> |    MC-dropout   |  11.15 |      &nbsp;&nbsp;4.09      |  78.46 |      1.23     | 0.19 |
> |    NatPN   |  9.32 |      &nbsp;&nbsp;5.30      |  78.45 |      1.62     | 0.34 |
> |    FGRM   | 8.52 |      &nbsp;&nbsp;6.97      |  79.23 |      1.77     | 0.48 |
>
>  > * Did the authors consider also papers such as those which propose learning heteroscedastic uncertainty with MC dropout as a baseline such as in Kendal et al.(https://arxiv.org/abs/1703.04977). This method is used quite significantly and it would have been nice to see how it performs in comparison.
>
> Reply: Thanks for the comment. We would like to draw your attention to Table 1 and Table 2 in the paper, where we have compared our method with MC dropout. The results show that our method outperforms MC dropout by a large margin across all the evaluation metrics.
>
>  > * It would be nice to see uncertainty maps of other models to compare. Is this possible?
>
> Reply: Thanks for the comment. Please kindly note that we have visualized the uncertainty maps of other models in Appendix Fig. 1. For clearer illustration, we have also updated Appendix Fig. 1 by including error maps of segmentation predictions (please refer to Fig. 2 of the uploaded PDF file). From the figure, we can see that our model generates uncertainty estimation maps that present better correlation with incorrect predictions when compared to other methods.
>
>  > * It is not entirely clear how the RL algorithm mitigates confidence miscalibration and OOD over confidence. This is only really mentioned in passing in Section 3.4 in the implementation details for Equation 1
>
> Reply: Thank you for the comment. For in-distribution (ID) calibration, our reinforcement learning (RL) algorithm aims to maximize the reward of Expected Calibration Error (ECE) metric. According to the formula of ECE metric shown in Appendix A.1, ECE metric quantifies the average disparity between confidence and accuracy, that is it effectively evaluates whether predictions characterized by higher confidence levels (indicative of lower uncertainty) are more likely to be accurate. By maximizing this metric with the RL algorithm, the confidence miscalibration can be reduced. For out-of-distribution (OOD) inference, our objective is to enable the model to assign higher uncertainty to the OOD regions compared with their ID counterparts. To achieve this, we have devised a reward function as the ratio between the uncertainty of OOD regions and that of ID regions. By maximizing this designed reward with the RL algorithm, the model learns to mitigate the OOD over confidence.
>
>  > * In the experiments, did the authors consider a model where only evidential learning was used as another baseline to compare against vanilla MLE? It would be nice to understand how strong of a baseline that is and whether this could be used in isolation.
>
> Reply: Thanks for the comment. Following your suggestion, we have included a model named “vanilla EDL”, in which only evidential learning is used as a comparison with vanilla MLE. The results in the tables below show that vanilla EDL achieves superior performance compared to vanilla MLE, demonstrating the benefits of evidential learning in uncertainty estimation. Moreover, our FGRM model further outperforms vanilla EDL, attributing to our RL framework with the designed fine-grained parameter update mechanism.
>
> **LC segmentation**
>
> |           |       | ID calibration |        | OOD inference |      |
> |-----------|-------|----------------|--------|---------------|------|
> |   Method  | ECE ↓ |      MI ↑      | Dice ↑ |      PR ↑     | BR ↑ |
> | Vanilla MLE | 19.46 |      2.88      |  71.22 |      0.88     | 0.08 |
> |    Vanilla EDL   |  12.98 |      4.20      |  73.66 |      1.45     | 0.21 |
> |    FGRM   |  9.63 |      5.87      |  74.88 |      1.85     | 0.47 |
>
> **ESD segmentation**
>
> |           |       | ID calibration |        | OOD inference |      |
> |-----------|-------|----------------|--------|---------------|------|
> |   Method | ECE ↓ |      MI ↑      | Dice ↑ |      PR ↑     | BR ↑ |
> | Vanilla MLE | 17.76 |      2.08      |  84.12 |      0.82     | 0.10 |
> |    Vanilla EDL   |  12.15 |      3.68      |  86.20 |      1.40     | 0.18 |
> |    FGRM   |  10.42 |      4.72      |  87.23 |      1.78     | 0.54 |

---

> > ### Comment · Reviewer_WcWT · 2023-08-18
> > **Response**
> >
> > Thank you for the detailed rebuttal and the additional experiments on i) Cityscapes and ii) results for vanilla EDL.
> >
> > I think this is a strong paper and the additional work provided by the authors strengthens the paper. My original score of 8 stands but I would defend publication of this manuscript.
> >
> > One note, the authors state in the rebuttal `Reply: Thanks for the comment. We would like to draw your attention to Table 1 and Table 2 in the paper, where we have compared our method with MC dropout. The results show that our method outperforms MC dropout by a large margin across all the evaluation metrics.`
> >
> > However, learned heteroscedastic uncertainty (https://arxiv.org/abs/1703.04977)) is not the same as MC dropout. In that paper, the authors develop a branched network (akin to hard-parameter sharing in multi-task learning) where one branch predicts the mean and the other the variance on a pixel-wise basis. This is fed into the loss function. MC-dropout can be used in addition to this method to estimate both epistemic and aleatoric uncertainty whereby the total uncertainty is the sum.

---

> > > ### Author Response · Authors · 2023-08-20
> > > **Thank You and Additional Experiments**
> > >
> > > Thank you for taking time to read our response and recognizing the strengths of our paper, along with the improvements from our additional work. We apologize that we initially misunderstood your suggested comparison method. Following your suggestion, we provide a comparison of our method with the learned heteroscedastic uncertainty (LHU, https://arxiv.org/abs/1703.04977) method in the tables below on LC and ESD segmentation datasets. Our method achieves superior performance on both ID and OOD scenarios, which demonstrates the effectiveness of our RL-based uncertainty estimation paradigm.
> > >
> > > **LC segmentation dataset**
> > > |            |        | ID calibration |         | &nbsp;&nbsp;&nbsp;&nbsp;&nbsp;&nbsp;&nbsp;&nbsp;&nbsp;&nbsp;&nbsp;&nbsp;&nbsp;&nbsp;&nbsp;&nbsp;&nbsp;&nbsp;OOD inference |       |
> > > |:---------:|:-----:|:--------------:|:------:|:-------------:|:----:|
> > > | Method | ECE ↓ | &nbsp;&nbsp;MI ↑ | Dice ↑ | PR ↑ | BR ↑ |
> > > | LHU | 17.14 | &nbsp;&nbsp;3.41 | 72.57 | 1.16 | 0.12 |
> > > | FGRM | 9.63 | &nbsp;&nbsp;5.87 | 74.88 | 1.85 | 0.47 |
> > >
> > > **ESD segmentation dataset**
> > > |            |        | ID calibration |         | &nbsp;&nbsp;&nbsp;&nbsp;&nbsp;&nbsp;&nbsp;&nbsp;&nbsp;&nbsp;&nbsp;&nbsp;&nbsp;&nbsp;&nbsp;&nbsp;&nbsp;&nbsp;OOD inference |       |
> > > |:---------:|:-----:|:--------------:|:------:|:-------------:|:----:|
> > > | Method | ECE ↓ | &nbsp;&nbsp;MI ↑ | Dice ↑ | PR ↑ | BR ↑ |
> > > | LHU | 16.62 | &nbsp;&nbsp;3.36 | 83.31 | 1.12 | 0.14 |
> > > | FGRM | 10.42 | &nbsp;&nbsp;4.72 | 87.23 | 1.78 | 0.54 |

---

### Official Review · Reviewer_DNkz · 2023-07-12

**Soundness:** 4 excellent
**Presentation:** 4 excellent
**Contribution:** 3 good
**Rating:** 5
**Confidence:** 5

**Summary:**

This paper empirically studied uncertainty estimation in safety-critical scene segmentation. The authors employed reinforcement learning (RL) methodologies, including fine-grained reward maximization (FGRM) framework and fisher information matrix for parameter updates. Additionally, to calibrate prediction risk and model confidence, the authors proposed a new reward function closely tied to uncertainty estimation. Through the experiments, the authors concluded the novel findings: (1) their method outperformed different types of state-of-the-art uncertainty estimation methods across all evaluation metrics; (2) the fine-grained parameter update mechanism improved the effectiveness of model tuning based on the reward function. Furthermore, the experimental results demonstrated the superiority of the proposed method on two medical datasets of safety-critical applications, specifically laparoscopic cholecystectomy scene segmentation and endoscopic submucosal dissection scene segmentation.

**Strengths:**

The reviewer significantly understands the significance and importance of the task proposed in this paper. The findings and methodologies can be significantly utilized in safety-critical scenarios such as medical applications. In addition, the use of reinforcement learning (RL) for uncertainty estimation in model tuning is a novel approach, and the proposed reward function specifically designed for uncertainty estimation is innovative. Furthermore, since the proposed method consistently outperformed state-of-the-art uncertainty estimation methods on evaluation metrics, the contribution of this paper is noteworthy. Additionally, the strengths of this paper can be summarized as follows:

1. The paper is easy to understand and easy to follow, making it accessible to a wide audience.

2. The authors clearly demonstrated the experimental results and effectively derived novel findings, supporting their claims.

3. The experimental results strongly support the effectiveness of the proposed method, highlighting its superiority over existing approaches.


**Weaknesses:**

The reviewer agrees that the task addressed in this paper has several strengths and is significantly relevant to the community, but this paper is not technically novel. The main reasons are listed below (See Limitations).

**Questions:**

1. The application scope is significantly limited (only applicable to endoscopy modality and only for medical imaging). It would be better if the proposed framework could be applied to a broader range of applications. It would be beneficial to have further discussions on this issue.



**Limitations:**

1. Limited novelty. The differences (novelty) compared to existing methods are not clearly demonstrated. In the rebuttal stage, the reviewer strongly hopes that the authors will emphasize the novelty of their proposed method and clearly highlight its differences from existing approaches, clarifying what is new and innovative.

2. Additionally, in the manuscript, the authors mention the inclusion of out-of-distribution (OOD) data alongside in-distribution data. The reviewer suggests justifying the use of OOD data in the experiments to strengthen the logical flow of the paper.

---

> ### Author Rebuttal · Authors · 2023-08-10
>
> Thank you for your positive comments on the significance and importance of the task tackled in our work, the novelty of using reinforcement learning (RL) in uncertainty estimation, and our strong experimental results. Our detailed responses to your comments are as follows.
>
> > * Applying the proposed framework to a broader range of applications.
>
> Reply: Thank you for your suggestions. To validate that our method can be applied to a diverse set of datasets, we further applied our method to the Cityscapes dataset for urban scene segmentation, which significantly differs from the surgical video datasets used in our paper. The Cityscapes dataset represents a diverse and realistic collection of urban scene images gathered from 50 cities with the semantic segmentation task involving 30 classes. The uncertainty estimation results on the Cityscapes dataset are shown in the table below. We can see that our method consistently outperforms all the comparison methods, indicating the potential of our method on the safety-critical autonomous driving scenarios.
>
> **Cityscapes dataset**
>
> |           |       | ID calibration |        | OOD inference |      |
> |-----------|-------|----------------|--------|---------------|------|
> |   Method&nbsp;&nbsp;&nbsp;&nbsp;&nbsp;&nbsp;  | ECE ↓ |      &nbsp;&nbsp;&nbsp;&nbsp;&nbsp;&nbsp;&nbsp;&nbsp;&nbsp;&nbsp;MI ↑      | Dice ↑&nbsp;&nbsp;&nbsp;&nbsp;&nbsp;&nbsp; |      PR ↑     | BR ↑ &nbsp;&nbsp;&nbsp;&nbsp;&nbsp;&nbsp;|
> | Backbone | 11.94 |      &nbsp;&nbsp;&nbsp;&nbsp;&nbsp;&nbsp;&nbsp;&nbsp;&nbsp;&nbsp;3.51      |  78.15 |      0.70     | 0.06 |
> |    LayerEnsemble   |  10.40 |      &nbsp;&nbsp;&nbsp;&nbsp;&nbsp;&nbsp;&nbsp;&nbsp;&nbsp;&nbsp;4.73      |  79.04 |      1.15     | 0.11 |
> |    DeepEnsemble   |  10.66 |      &nbsp;&nbsp;&nbsp;&nbsp;&nbsp;&nbsp;&nbsp;&nbsp;&nbsp;&nbsp;4.91      |  79.10 |      1.37     | 0.26 |
> |    LDU   |  9.14 |      &nbsp;&nbsp;&nbsp;&nbsp;&nbsp;&nbsp;&nbsp;&nbsp;&nbsp;&nbsp;5.75      |  78.69 |      1.55     | 0.35 |
> |    MC-dropout   |  11.15 |      &nbsp;&nbsp;&nbsp;&nbsp;&nbsp;&nbsp;&nbsp;&nbsp;&nbsp;&nbsp;4.09      |  78.46 |      1.23     | 0.19 |
> |    NatPN   |  9.32 |      &nbsp;&nbsp;&nbsp;&nbsp;&nbsp;&nbsp;&nbsp;&nbsp;&nbsp;&nbsp;5.30      |  78.45 |      1.62     | 0.34 |
> |    FGRM   | 8.52 |      &nbsp;&nbsp;&nbsp;&nbsp;&nbsp;&nbsp;&nbsp;&nbsp;&nbsp;&nbsp;6.97      |  79.23 |      1.77     | 0.48 |
>
> >  * The differences (novelty) compared to existing methods are not clearly demonstrated.
>
> Reply: Thank you for your comments. We would like to highlight the novelty of our proposed method based on the following three key aspects.
>
> i. Our proposed FGRM framework is **the first work** to directly optimize uncertainty estimation reward function in a RL paradigm for effective uncertainty estimation. Existing uncertainty estimation methods are based on standard supervised learning with task-specific objectives, such as cross-entropy loss or Dice loss, which are not directly related to the uncertainty estimation task. Instead, our method leverages the RL paradigm, which enables us to optimize a reward function closely tied to uncertainty estimation metrics for explicit model tuning of uncertainty estimation. This is not feasible in standard supervised learning due to the discrete and non-differentiable nature of these metrics, which are essential for effective back propagation. Such framework novelty is also acknowledged by reviewer WcWT.
>
> ii. We design a novel fine-grained parameter update mechanism based on fisher information matrix for the reward maximization in RL. Existing works on RL in vision tasks typically adopt the policy gradient theorem to estimate the gradient of the reward term, in which the gradient of each parameter is weighted by the reward uniformly. Instead, our designed mechanism imposes fine-grained reward-weighting for each network parameter individually so as to enable more effective model tuning based on the reward function. Such fine-grained parameter update design novelty is also acknowledged by reviewer WcWT and reviewer iwgk.
>
> iii. We leverage evidential deep learning for the pre-training of the segmentation backbone. Evidential learning explicitly parameterizes the conjugate prior of the categorical distribution, enabling us to quantify the aleatoric uncertainty and epistemic uncertainty separately. This is in contrast to standard segmentation models, where the probability values from the softmax layer mix both types of uncertainty together. The novelty of incorporating evidential learning is also acknowledged by reviewer WcWT and reviewer mEtW.
>
> Based on the three points mentioned above, we believe that our work brings new insights and valuable contributions to the task of uncertainty estimation, which would be of great interest to the research community.
>
>  > * Additionally, in the manuscript, the authors mention the inclusion of out-of-distribution (OOD) data alongside in-distribution data. The reviewer suggests justifying the use of OOD data in the experiments to strengthen the logical flow of the paper.
>
> Reply: Thanks for your suggestion. The logic to include OOD data in our experiments is to provide a comprehensive evaluation of our proposed uncertainty estimation method. In practical deployment scenarios, segmentation models often encounter data that deviates from their training distribution, resulting in potentially inaccurate segmentation predictions. By including OOD data, we aim to simulate these real-world challenges and validate the effectiveness of our method when dealing with OOD data, which is crucial for reliable uncertainty estimation.

---

### Official Review · Reviewer_mEtW · 2023-07-27

**Soundness:** 3 good
**Presentation:** 2 fair
**Contribution:** 3 good
**Rating:** 7
**Confidence:** 2

**Summary:**

This paper introduces a new uncertainty estimation method for medical imaging segmentation tasks. The method relies on a pre-trained segmentation network that uses evidential learning to produce the parameters of a Dirichlet distribution over the class probabilities, which can be translated to aleatoric and epistemic uncertainty estimates. Then, an uncertainty estimation network is trained by updating its parameters based on a custom reward function weighted by the Fisher information matrix for all of the parameters. Experiments on two medical image segmentation datasets show that the method outperforms baselines including ensemble-based methods, deterministic uncertainty estimation methods, dropout, and an evidence update method. Ablations show the value of each part of the method.

[Note: I do not work on uncertainty estimation or reinforcement learning, so other reviewers may be more qualified than me to comment here, and it is definitely possible I have misunderstood some of the paper.]

**Strengths:**

- The paper contains an extremely thorough ablation study and baseline comparison which demonstrate the value of each component of the system and a clear benefit over baselines (to be clear, this is a very big strength).
- The idea of incorporating the outputs of evidential learning in the segmentation output into the training of an uncertainty estimation network seems interesting and is to my knowledge (from a brief literature search -- I am not familiar with this field) novel.

**Weaknesses:**

- **I do not understand how this is a reinforcement learning strategy, as opposed to standard training of another neural network that predicts uncertainty.** In particular, the form of the “RL Reward Maximization” part of Algorithm 1 looks exactly like standard training where network parameters are being updated via a loss function, and as far as I can tell, there is no notion of an “agent” performing actions in sequential time steps. I still think there is a contribution here, because the empirical results are clearly improved over the baselines, and there may be a misunderstanding on my part because I am not a reinforcement learning researcher. However, I think the paper may need a significant rewrite to clarify it and fully meet the bar for acceptance: either the paper should be written with the contribution being essentially a novel loss function, or the exposition needs some clarification in framing this as a reinforcement learning problem, as I did spend a good amount of time trying to understand this and could not figure it out. I am happy to update my rating later based on subsequent discussion on this point.
- **Some details of the ablation study are unclear.** It appears evidential learning was added after reward maximization. How was reward maximization performed without the aleatoric and epistemic uncertainties provided from the evidential learning task (i.e. was R computed as in Appendix A.1? If so, how?)?
- **OOD tasks are not realistic.** I do not think the OOD perturbations induced by Hendrycks et al are a realistic approximation to the types of OOD examples that would be seen in a medical segmentation task, where one would be especially concerned with unusual anatomy/pathology, etc. I don’t consider this a dealbreaker for accepting this paper, however, as it certainly seems reasonable that these OOD perturbations help in formulating the reward function; I am just less able to draw conclusions from the “OOD Inference” columns in the results tables.
- **Visual results are hard to interpret.** Figure 2 does not provide the ground truth segmentations, so it is hard to tell where the model is incorrect, especially for readers unfamiliar with these types of images. Figure 1 in Appendix A.4 does show ground truth segmentations, but it would be helpful to see an error map of the segmentations to compare with the uncertainty.

**Questions:**

- Why is this considered a reinforcement learning algorithm?
- How was reward maximization performed without the aleatoric and epistemic uncertainties provided from the evidential learning task?

**Limitations:**

The paper has a clear “Limitations” section which describes a desire to unify the framework for dealing with in-distribution and out-of-distribution examples. I would like to see some additional discussion of the fact that that the OOD evaluations are not fully representative of what might be encountered in medical image segmentation tasks.

---

> ### Author Rebuttal · Authors · 2023-08-10
>
> We sincerely thank you for your positive comments on our thorough ablation study, clear benefits over baselines, and novel methodology. We would like to address each of your questions below.
>
>  > * Why is this considered a reinforcement learning algorithm?
>
> Reply: Thank you for your comment. Our method aims to explicitly optimize uncertainty estimation metrics as a reward function to directly calibrate prediction risk and model confidence. However, achieving this in standard supervised learning is unfeasible due to the inherent discrete and non-differentiable characteristics of these metrics, which are necessary for effective back propagation. To overcome this limitation, our approach formulates the uncertainty estimation task as a reinforcement learning (RL) paradigm. This allows us to directly optimize uncertainty estimation metrics, like the Expected Calibration Error, that are both discrete and non-differentiable. This intricate process involves the formulation of a Markov Decision Process (MDP), where the segmentation model takes on the role of the policy. Within this framework, segmentation predictions are actions and the input image is the state, as elaborated in a recent work on tuning computer vision models with task rewards [31]. The policy performs the action (segmentation) with a single time step, thus there is no notion of sequential actions across time. The “RL Reward Maximization” part in Algorithm 1 actually differs from standard training in terms of feedback signal and policy optimization. In RL reward maximization, feedback is received in the form of rewards based on the performed action. The objective is to learn a policy that maximizes the rewards, which are usually discrete and non-differentiable. In standard training, the objective involves minimizing a loss function based on labeled data, which is continuous and differentiable. Furthermore, RL reward maximization typically involves optimizing a policy that maps states to actions, aiming to maximize rewards. Standard training focuses on optimizing parameters of a model to minimize a specific loss function. We appreciate your feedback and will provide a clearer explanation of our RL paradigm in the paper.
>
>  > * For the ablation study, how was reward maximization performed without the aleatoric and epistemic uncertainties provided from the evidential learning task?
>
> Reply: For the ablation study, when evidential learning is not employed, it means that a softmax layer instead of an evidence layer is used in the segmentation backbone. Since the softmax layer cannot separate the aleatoric and epistemic uncertainties, following the baseline setting in previous methods [10], the maximum class probability (MCP) generated by the softmax function is used as both the aleatoric and epistemic uncertainty. This MCP-based uncertainty is subsequently used in the computation of the uncertainty-related reward function, which is maximized with our fine-grained parameter update RL algorithm.
>
> > * The OOD perturbations are not a realistic approximation to the types of OOD examples that would be seen in a medical segmentation task.
>
> Reply: Thanks for your comment. For the generation of out-of-distribution (OOD) examples, we randomly applied one out of four types of perturbations, including contrast, brightness, pixelate, and noise, to the original images. Among these, the perturbations of contrast, brightness, and pixelate are likely to be encountered in surgical video data, due to variations in the surgical environment. Some examples of generated OOD examples are presented in Fig. 3(a) of the uploaded PDF file. We can see that those perturbations mainly adjust the contrast, brightness, and clarity of surgical images, which might be encountered in practical scenarios. However, we acknowledge the reviewer’s suggestion that another important type of OOD cases for surgical video data would involve different anatomical structures or pathologies. We thus conducted additional tests on endoscopic submucosal dissection (ESD) surgical data, which involves organs distinct from those covered by our training data. Specifically, we applied the model trained on ESD surgical data of the stomach to test on ESD surgical data of the esophagus and the rectum. The uncertainty estimation results for these OOD cases are shown in Fig. 1(b) of the uploaded PDF file accompanying our global response. The figure demonstrates that our method effectively provides meaningful uncertainty estimation for OOD cases involving different organs.
>
> > * It would be helpful to include ground truth segmentations in Figure 2 and error maps of segmentations in Figure 1 in Appendix A.4.
>
> Reply: Thank you for your valuable suggestions. We have included the ground truth segmentations and error maps in Fig. 2. and error maps in Fig. 1 in Appendix A.4. The updated figures can be visualized in Fig. 1 and Fig. 2 of the uploaded PDF file. In Fig. 1, we can observe that our estimated uncertainty maps effectively highlight regions where the segmentation predictions are unreliable. For example, the bottom right areas in the first and second columns of Fig. 1 are incorrectly segmented, and our model assigns high uncertainty values to these regions accordingly. In Fig. 2, we can see that our model generates uncertainty estimation maps that present better correlation with incorrect predictions compared to other methods.

---

> > ### Comment · Reviewer_mEtW · 2023-08-20
> > **Terminology questions remain, but I have raised my score.**
> >
> > Thanks to the authors for their thorough responses to my and other reviews.
> >
> > Thanks to the authors for highlighting the connection to reference [31] and clarifying their view of the reinforcement learning framework. The rebuttal states: “segmentation predictions are actions and the input image is the state” — to me, this still doesn’t seem like an RL framework, because the actions do not affect the state of the system. However, it seems to me that [31] is characterized by a similar issue, and reframing the contribution may make the paper much easier to understand by future readers. I do think this is a good paper that should be accepted — at this point we are just discussing a terminology difference. I will defer to the AC and other reviewers on this point and will raise my score.
> >
> > I am satisfied with the response to all other points in my initial review, and especially appreciate the additional tests on endoscopic submucosal dissection (ESD) surgical data and improved visualizations of the results.
> >
> > I have also been following the discussion with reviewer jxwz about comparison to other medical imaging segmentation baselines. While the original paper already contained a thorough empirical characterization of the results, I think the latest response in the rebuttal further strengthens the paper and I am satisfied with the response on that point.

---

> > > ### Author Response · Authors · 2023-08-21
> > > **Thank You for Your Response**
> > >
> > > Thank you for the support of our work and we are glad to see that you are satisfied with our responses. Regarding the RL framework terminology, we intended to align with reference [31], but we also understand and agree with your point that RL might not be too precise in our context. We would like to rephrase our uncertainty estimation problem as a reward optimization process which is solved using the RL algorithms, rather than formulating a RL system. This writing description will be clarified in the final version. Thanks again for the rigorous suggestion.

---

### Author Rebuttal · Authors · 2023-08-10

We appreciate the reviewers for taking their time to review and provide constructive feedback. We are glad to see that most reviewers recognized the novelty of our method, the strength of our experimental evaluations, and the good presentation of our paper.

We have made every effort to address all the concerns raised, including conducting additional experiments on the Cityscapes dataset to validate that our method can have broad usage, adding more comparison experiments and ablation studies, and providing clarifications where needed. Our detailed responses have been provided to each reviewer separately in below. We have also uploaded a PDF file containing updated figures for a clearer visualization of uncertainty estimation and more ablation results.

We hope that our rebuttal can successfully address the reviewers' questions, and look forward to receiving the reviewers’ support for our work.

---

### Decision · Program_Chairs · 2023-09-21

**Decision:**

Accept (poster)

**Comment:**

This paper introduces a new uncertainty estimation method for medical imaging segmentation tasks.
All but one reviewer recommended for acceptance. The key strengths are great paper writing and many novel contributions supported by thorough experimental results. Rw jxwz is the only one recommended borderline rejection. jxwz's main concerns are the lack of comparison with auxiliary network-based methods and SoTA medical image methods. The authors provided additional experimental results addressing these concerns. In the end, Rw jxwz didn't update the rating, but Rw mEtW followed the discussion and mentioned that the additional results further strengthened the paper. AC agrees with most of the reviewers and recommends acceptance.